# LR0.FM: Low-Res Benchmark and Improving robustness for Zero-Shot Classification in Foundation Models

**Priyank Pathak[1], Shyam Marjit[2], Shruti Vyas[1] & Yogesh S Rawat[1]**
[1]University of Central Florida,    [2]IIIT Guwahati
`priyank@ucf.edu, shyam.marjit@iiitg.ac.in, {shruti, yogesh}@ucf.edu`
https://ucf-crcv.github.io/lr0.fm

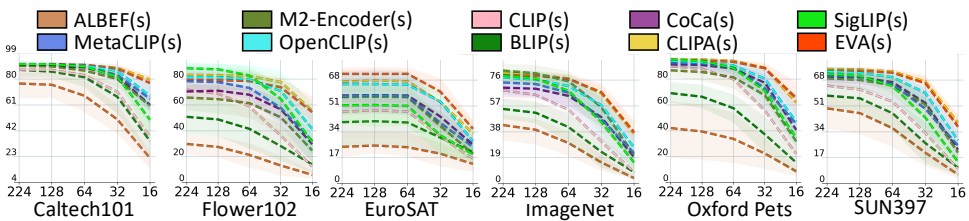

Figure 1: **Top-1 zero-shot classification accuracy (y-axis) vs resolution (x-axis)**: Backbones for foundation models are merged as shade, with average performance across backbones in the dark.

## Abstract

Visual-language foundation Models (FMs) exhibit remarkable zero-shot generalization across diverse tasks, largely attributed to extensive pre-training on large-scale datasets. However, their robustness on low-resolution/pixelated (LR) images, a common challenge in real-world scenarios, remains underexplored. We introduce **LR0.FM**, a comprehensive benchmark evaluating the impact of low resolution on the zero-shot classification performance of *10* FM(s) across *66* backbones and *15* datasets. We propose a novel metric, **Weighted Aggregated Robustness**, to address the limitations of existing metrics and better evaluate model performance across resolutions and datasets. Our key findings show that: (i) model size positively correlates with robustness to resolution degradation, (ii) pre-training dataset quality is more important than its size, and (iii) fine-tuned and higher resolution models are less robust against LR. Our analysis further reveals that the model makes semantically reasonable predictions at LR, and the lack of fine-grained details in input adversely impacts the model's initial layers more than the deeper layers. We use these insights and introduce a simple strategy, **LR-TK0**, to enhance the robustness of models without compromising their pre-trained weights. We demonstrate the effectiveness of **LR-TK0** for robustness against low-resolution across several datasets and its generalization capability across backbones and other approaches. *Code is available at this link.*

## 1 Introduction

Vision-Language Foundation Models (FMs), such as CLIP (Radford et al., 2021), LLaMA (Touvron et al., 2023), and other variants, have shown extraordinary generalization capabilities across a wide range of downstream tasks, including image classification (Ilharco et al., 2021), object detection (Zhong et al., 2022), and semantic segmentation (Xu et al., 2023). These models benefit from large-scale, multi-modal pre-training on diverse datasets like DataComp-1B (Gadre et al., 2023) and LAION-5B (Schuhmann et al., 2022), enabling them with zero-shot capabilities. Although these models excel on high-resolution benchmarks, their performance with low-resolution (LR) pixelated images, a common real-world challenge, remains adequately underexplored.

Low-resolution images frequently arise in various practical scenarios, such as surveillance footage (Davila et al., 2023), satellite imagery (Patil et al., 2017), and privacy-protected pixelated

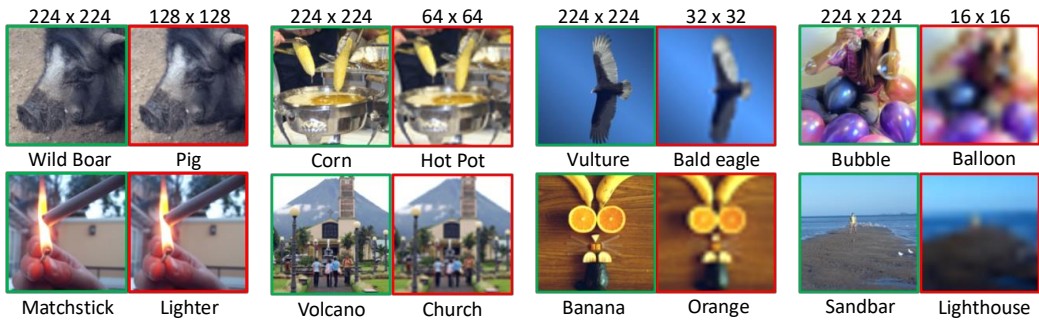

Figure 2: **Zero-Shot misclassifications:** EVA-CLIP [2023a] correct classification at 224×224 (green) & misclassification at lower resolution (red). However, ImageNet labels-based mispredictions are semantically reasonable (humans), indicating viability of pre-trained weights at low resolution.

data (Zhou et al., 2020) *etc.* In these cases, details crucial for accurate classification may be obscured by artifacts like pixelation and compression, leading to substantial performance degradation. For instance, small objects (faces) within larger images (Cheng et al., 2019) pose unique challenges, often requiring models to rely on limited visual cues. Given the widespread presence of LR images in real-world applications, it is crucial to understand how robust FMs are in these settings. 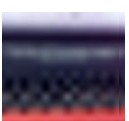

Motivated by this, we present an in-depth benchmarking study of FMs, focusing on their zero-shot classification performance under LR conditions. We introduce LR0.FM, a comprehensive benchmark that evaluates **10 foundation models** across **66 backbones** and **15** diverse image classification **datasets**, ranging from large-scale datasets like ImageNet (Deng et al., 2009) to fine-grained and texture-specific datasets like Oxford Pets (Parkhi et al., 2012) and DTD (Cimpoi et al., 2014). Our study systematically examines the effects of resolution degradation, revealing key insights into how model size, pre-training dataset quality, and fine-tuning impact robustness in LR scenarios.

Metrics for measuring robustness $\gamma$ (Schiappa et al. (2024b)) and its averaging across datasets (SAR) have some limitations; 1) They can produce misleadingly high scores when models perform poorly on challenging datasets, and 2) They tend to ignore certain datasets, skewing the overall comparison. To address these, we propose a new metric, **Weighted Aggregated Robustness (WAR)**, which provides a more balanced evaluation by considering performance drops across datasets more fairly.

Our analysis reveals several interesting insights. Larger models tend to maintain robustness better when faced with LR inputs, while the quality of the pre-training dataset is more crucial than its size in preserving performance. Furthermore, fine-tuned models and those with higher-resolution inputs significantly underperform against resolution drop. We also observe that although models struggle at low-resolution (fig. 1) and loss of fine-grained details (fig. 2: *e.g.* Vulture vs Bald Eagle, Bubble vs Balloon *etc.*), their predictions often remain semantically reasonable, even at extreme resolutions (fig. 2: *e.g.* Orange vs Banana, Church vs Volcano *etc.*). *Supplementary* demonstrates more examples (including real-world) where such mispredictions are made. This suggests a solution for low-resolution does not require extensive modifications to the model and its pre-trained weights.

Based on these insights, we propose a simple yet effective solution, **LR-TK0: LR-Zero-Shot Tokens**, which introduces low-resolution-specific tokens to enhance robustness without altering the pre-trained model weights. Our method preserves the model's semantic reasoning capabilities while compensating for the loss of fine-grained detail, offering a feature super-resolution-like approach (Chen et al., 2024). By training on synthetic diffusion-based high-resolution images, LR-TK0 improves performance in low-resolution zero-shot classification tasks, making FMs more robust for practical, real-world applications.

In summary, we make the following contributions in this work,

1. We present **LR0.FM**, a comprehensive benchmarking of Visual-Language Foundation Models (FMs) on zero-shot classification of low-resolution images, providing several key insights. To the best of our knowledge, no prior work has explored this aspect of FMs.
2. We introduce a simple and effective method, **LR-TK0**, to enhance model robustness against low-resolution inputs without altering the pre-trained weights.
3. We introduce Weighted Aggregated Robustness (**WAR**), a novel robustness metric for evaluating models under challenging conditions, overcoming the limitations of existing metrics.

Table 1: **Benchmark Models (66 Backbones):** Pre-training is image-text pairs from datasets like DataComp-1B (DC-1B) (Gadre et al., 2023), Conceptual Captions (CC) (Sharma et al., 2018), Conceptual 12M (C-12M) (Changpinyo et al., 2021). Text Encoders are mostly modified vanilla transformers (Tran.)(Vaswani et al., 2017). Vision backbones use (modified) ViTs (Dosovitskiy, 2021).

| Models | #Backbones | Pre-training (Dataset / Size Billion:B & Million:M) | | Text Encoder |
|---|---|---|---|---|
| CLIP [2021] | 4 ViTs & 5 ResNets | WIT-400M [2021] | 400M | Tran. [2019] |
| OpenCLIP [2021] | 8 ViTs | DC-1B, LAION-2B[2022], DFN-5B[2023] | 1B-5B | Tran.[2021] |
| MetaCLIP [2024a] | 8 ViTs | *Self* | 400M-2.5B | OpenCLIP |
| CLIPA (v1&v2) [2023b; 2023c] | 7 ViTs | DC-1B, LAION-2B [2022] | 1B-2B | Autoregressive Tran. [2017] |
| SigLIP [2023] | 8 ViTs | WebLI [2023b] | 10B | Tran. |
| CoCa [2022] | 3 ViTs | LAION-2B [2022], COCO [2014] | 2B | Tran. Decoder |
| $M^2$-Encoder[2024] | $3M^2$-Encoder | BM-6B [2024] | 6B | Magneto [2023] |
| ALBEF [2021] | 4 ALBEF (ViT) | COCO [2014], Visual Genome [2017], CC, SBU Captions [2011], C-12M | 4M-14M | BERT [2019] |
| BLIP [2022b] | 8 ViTs | ALBEF [2021], LAION-400M [2021] | 14M-129M | BERT [2019] |
| EVA-CLIP(&18B) [2023a; 2023b] | 8 EVA(s) (ViT(s)) | LAION-400M[2021], LAION-2B[2022], Merged-2B [2023a] | 400M-2B | OpenCLIP |

## 2 RELATED WORKS

**Foundation Models (FM):** Large-scale models (Kirillov et al., 2023; Girdhar et al., 2023), pre-trained on massive datasets, demonstrate generalization across numerous downstream tasks. For example, CLIP (Radford et al., 2021) embeds ∼400 million image-text pairs in a shared feature space for zero-shot image classification and image-text retrieval. It is also effective in other domains like video-text retrieval (Luo et al., 2022), and video and audio understanding (Lin et al., 2022; Guzhov et al., 2022). Joint vision-text learning has also succeeded in tasks such as self-supervision (Miech et al., 2020), few-shot (Alayrac et al., 2022), multi-modal retrieval (Yu et al., 2022) *etc.* However, the robustness of these models against *real-world challenges e.g.* harmful images (Qu et al., 2024), image quality (Wu et al., 2024), text quality (Xu et al., 2024b), *etc.* requires further exploration.

**Zero Shot:** Zero Shot/Open-set/In-the-wild image classification predicts an unseen class by matching the image with labels (Sun et al., 2023a). In the past, traditional models have been tested for their zero-shot capabilities (Chao et al., 2016; Xian et al., 2017), however, FMs are better suited for this task. Benchmarking their zero-shot capabilities is a relatively newer area of research (Schiappa et al., 2024a; Schulter et al., 2023). To assess the performance comprehensively, we have expanded the pool of models from traditional 10-11 FM backbones *e.g.* 4 backbones (Li et al., 2022a), 9 backbones (Liu et al., 2024), 6 backbones ((Zhang et al., 2024)) *etc.* to *66 backbones*.

**Low Resolution (LR):** LR images are captured in various practical scenarios and are sometimes used intentionally for computational cost reduction (RECLIP (Li et al., 2023a)). LR benchmarks mostly focus on face recognition (Luevano et al., 2021; Li et al., 2018; Tirupattur et al., 2021), with some work in zero-shot/unconstrained recognition (Li et al., 2019; Cheng et al., 2019). Super Resolution (Ohtani et al., 2024; Gao et al., 2023) are often domain-specific or restores only $\geq 64 \times 64$. However, there is a lack of study on the robustness of FM(s) against real-world challenges (Xu et al., 2024b; Schiappa et al., 2023), with no previous work on very LR. We benchmark FM(s) against LR images and propose a lightweight solution for improving robustness, without training on any of the target datasets (Chen et al., 2024).

## 3 BENCHMARKING SETUP

**Model:** Table 1 lists all 10 Foundation models used in our benchmarking[1]. CLIP, OpenCLIP, Meta-CLIP, CLIPA, and SigLIP use the same ViT model with different pre-training datasets and slight architectural modifications (*e.g.* layer norm position, token masking *etc.*). $M^2$-Encoder (built on top of CoCa), ALBEF, and BLIP use modified cross attention between text and vision transformers. EVA-CLIP is a family of models equipped with recent advancements *e.g.* architectural modifica-

---

[1]EVA-CLIP (Sun et al., 2023a) & EVA-CLIP-18B (Sun et al., 2023b) merged into one.

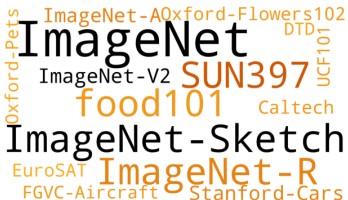 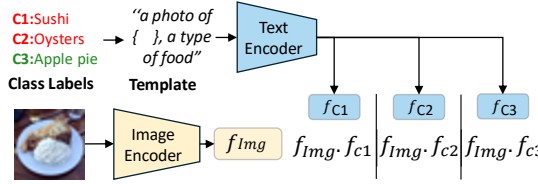

Figure 3: *Left*: **Dataset**: Size $\propto \log$ # test images, and color gradient $\propto$ # of test classes **orange** is 10 & **black** is 1000 classes). *Right*: **Zero Shot Evaluation**: Food-101 image ($32\times32$) generates image embeddings $f_{Img}$, while class labels are filled in templates (1 shown) generating text embeddings (averaged across templates). The dot product of $f_{Img}$ with text features gives classification logits.

tions, token dropping, training via distillation *etc.* surpassing all existing works. Backbones are referred to using their publicly available pre-trained weights, *e.g.* CLIP-ViT L (400M), which means: CLIP model ViT-L architecture, pre-trained on 400 million datasets. 'B' would indicate a billion.

**Dataset:** Figure 3 (left) highlights benchmarking datasets size and the number of classes for: Imagenet [2009], ImageNet-A [2021b], ImageNet-V2 [2019], ImageNet-R [2021a], ImageNet-Sketch (ImageNet-SK) [2019], Caltech101 [2007], DTD split-1 (DTD) [2014], Food101 [2014], SUN397 [2014] Stanford Cars (Cars) [2020], FGVC Aircraft (Aircraft) [2013], Oxford Pets (Pets) [2012], Oxford Flowers102 (Flowers102) [2016], EuroSAT [2019], UCF101 [2012]. Details in *Supplementary*.

**Zero-Shot Image Classification** We adopt CLIP (Radford et al., 2021) evaluation protocol for all the models as shown in fig. 3 (right). Image encoder generates embeddings for images, while test labels are used with dataset-specific templates (multiple templates, *Supplementary*) *e.g.* "a photo of a [label]". Model's Text encoder generates final text embeddings (averaged across all templates) for the class label. The dot product of visual and text embeddings produces class logits, with the highest logit score determining the predicted class. Accuracy is computed using Top-1 match.

**Low Resolution:** Models are evaluated on their pre-trained resolution, namely $224\times224$ $256\times256$, $378\times378$ *etc.* Low resolution is simulated by downsampling HR images to $16\times16$, $32\times32$, $64\times64$, and $128\times128$ using bicubic interpolation, followed by model specific preprocessing similar to their HR counterparts, *e.g.* resizing to $224\times224$, center crop, *etc.* Performance degradation starts below $64\times64$ (fig. 1), so we focus mainly on $16\times16$ and $32\times32$. This downsampling mimics pixelation as seen in low-resolution cameras (*e.g.* self-driving cars) and distant images (*e.g.* CCTV), *etc.*

**Evaluation Metrics**: We represent top-1 accuracy on the dataset '$D$' with a resolution $n\times n$ as $A_n^D \in [0,1]$, *e.g.* HR accuracy $A_{HR}^D \geq A_n^D$ (LR accuracy), where HR is model specific $\in \{224, 256, 372, 384, 512\}$. Top-1 scores averaged across datasets is **ACC-n**. Robustness against artifacts (Schiappa et al., 2024b) is measured by relative robustness ($\gamma_n^D = 1 - (A_{HR}^D - A_n^D)/A_{HR}^D$). $\gamma_n^D$ is dataset-specific, and it is common to average scores across datasets for model comparison, denoted by **Simple Aggregated Robustness (SAR-n)**. *Higher number indicates more robustness*. However, there are two significant issues with $\gamma_n^D$ and SAR-n:

***Problem A) Misleading high robustness:*** If the model performs poorly on a challenging dataset *i.e.* performance close to random predictions, then downsampling will likely maintain this random prediction with minimal drop in accuracy, giving abnormally high robustness score. *Ex.* 'ALBEF (4M)' for Aircraft dataset, ($A_{rand}^{aircraft}=1\%$), $A_{HR}^{aircraft}=2.7\%$, $A_{16}^{aircraft}=1\%$, $\gamma_{16}^{aircraft}=37\%$, *i.e.* random predictions yields $\sim$40% robustness (40% robustness is among the highest, more in *Supplementary*).

***Solution: Improved Relative Robustness*** $\Gamma_n^D$**:** A naive solution is to calculate relative robustness only for correct predictions at the HR resolution. However, tracking predictions for each model across all datasets might not be scalable, especially if the dataset contains millions of images. We propose *zero-ing out robustness near random predictions*. We first define *accuracy gap* for the model on a dataset with '$C$' classes as $\mathcal{E}_D = A_{HR}^D - A_{rand}^D$, with $\mathcal{E}_D \in [0,1]$, and $A_{rand}^D = 1/C$ represents random prediction accuracy[2]. If $A_{HR}^D >> A_{rand}^D$, $\mathcal{E}_D$ will be high. Conversely, if $A_{HR}^D \simeq$ random prediction, $\mathcal{E}_D \to 0$. Using $\mathcal{E}_D$, we compute **improved relative robustness** $\Gamma_n^D$ as

$$\Gamma_n^D = \gamma_n^D \times (1 - e^{-\alpha(\mathcal{E}_D)^2}) \quad | \quad \alpha >> 1 \quad \& \quad 0 \leq \mathcal{E}_D \leq 1 \tag{1}$$

---

[2]Random guessing one of the 'C' class yields $1/C$ accuracy, referred to as $A_{rand}^D$ in this work.

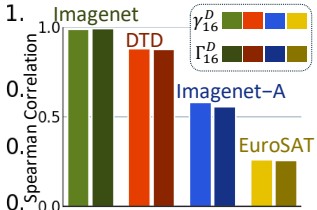 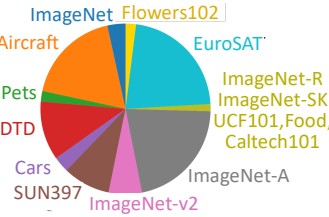

Figure 4: *Left*: **Improved $\Gamma_n^D$ vs traditional $\gamma_n^D$**: $\Gamma_n^D \approx \gamma_n^D$ except near random predictions ($\mathcal{E}_D \rightarrow 0$). *Mid*: **Correlation** between the ordering of models after averaging of robustness (SAR) across datasets ($\gamma_{16}^D$ & $\Gamma_{16}^D$) with dataset's true ordering. SAR final ranking ignores datasets like EuroSAT (0.26). *Right*: **Optimized dataset weights** for WAR-16. *Supplementary* contains numeric value.

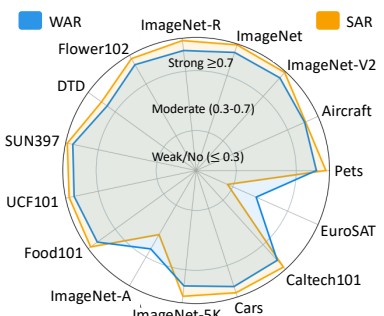 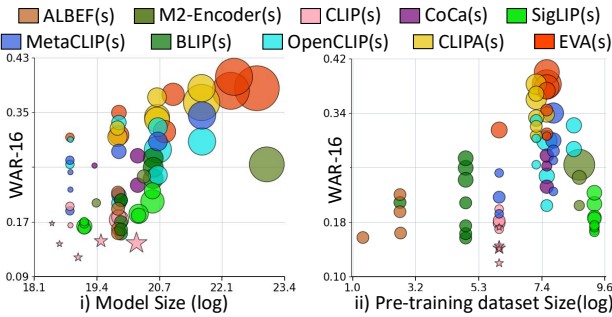

Figure 5: Evaluations at $16 \times 16$. *Left:* **SAR vs WAR:** WAR improves the correlation (between the ordering of models after aggregation with individual datasets) for EuroSAT ($0.26 \rightarrow 0.49$ and ImageNet-A ($0.56 \rightarrow 0.68$), both computed via $\Gamma_{16}^D$. *Right*: **i) Model Size & ii) Pre-training dataset size** positively impacts robustness. (i) Dot size $\propto$ GFLOPs, no impact on robustness (ii) Dot size $\propto$ Model Size, positively impact robustness. ResNets ($\star$), and transformers ($\bigcirc$).

when $\mathcal{E}_D \sim 0$ *i.e.* near random predictions, $\Gamma_n^D \sim 0$, otherwise $\Gamma_n^D \approx \gamma_n^D$, as shown in fig. 4 (left). Hyperparameter $\alpha$ is the rate at which $\Gamma_n^D$ declines as accuracy approaches random prediction. We chose $\alpha = 200$ as a middle between 100 (the drop at $\mathcal{E}_D \sim 0.2$) and 500 (the drop at $\mathcal{E}_D \sim 0$).

***Problem B) SAR overlooks datasets:*** When comparing models, their robustness scores are averaged across datasets (SAR). Ideally, model ranking, after averaging robustness across datasets, should stay consistent with their rankings on individual datasets. However, fig. 4 (mid) shows the rankings of 66 models after averaging correlate (Spearman Rank) highly with ImageNet (0.99) and DTD (0.88), but only moderately with ImageNet-A (0.56) and weakly / not with EuroSAT (0.26). Most datasets follow the ImageNet trend, influencing the final model rankings and minimizing the impact of datasets like ImageNet-A and EuroSAT (behave differently) as if these datasets aren't present.

***Solution: Weighted Aggregated Robustness:*** Averaging the robustness scores gives each dataset score of 1. We propose adjusting the dataset weights so that the *model rankings after aggregation reflect each dataset fairly* (fig. 4 (right)). Weights are optimized such that the correlation (Spearman) between the model rankings after the weighted average and individual dataset rankings are maximized. The weighted sum of robustness is: **WAR-n**$= \sum_d^{\text{Datasets}} |\Gamma_n^d \times w_n^d| / \sum_d^{\text{Datasets}} |w_n^d|$, where $w_n^d$ is dataset weight, and $\Gamma_n^d$ is dataset-specific improved robustness score for the resolution $n \times n$.

We use Ax tool (Bakshy et al., 2018) for optimizing the weights of the dataset $w_{16}^d \in [0.1, 1]$ such that the Spearman correlation (SC) between the final model ranking obtained after the weighted averaging and individual dataset ranking is maximized on empirically found (more in *Supplementary*):

$$0.95 \times \big(\text{SC(Imagenet)} + \text{SC(ImageNet-V2)} + \text{SC(DTD)}\big) + \text{SC(ImageNet-A)} + \text{SC(EuroSAT)} \quad (2)$$

Optimizing $w_{16}^d$ may give minimal weights to some datasets, thus WAR-n may not reflect the true robustness and is more apt for model comparisons, representing all the datasets. Hence we use both Weighted Aggregated Robustness (WAR) using improved relative robustness $\Gamma_n^D$ (eq. (1)) and simple averaging (SAR) using traditional robustness $\gamma_n^D$ for evaluating models. *Note*, $\gamma_n^D$ **and** $\Gamma_n^D$ **measure dataset robustness while SAR and WAR measure averaged robustness across the datasets**.

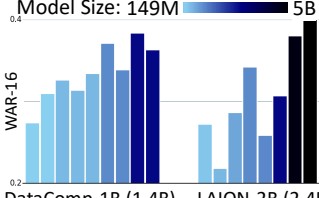 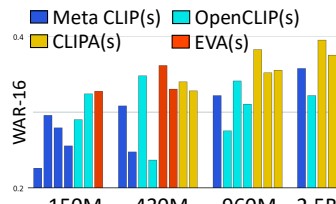 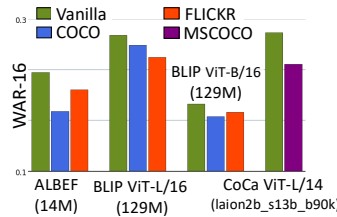

Figure 6: *Left:* **DataComp-1B vs LAION-2B**: Smaller DataComp-1B pre-training helps robustness. Models are ordered via size. *Mid:* **Model comparison w/o Size**: Models binned into size buckets ($\pm$30M). *Right:* **Fine-tuning degrades** robustness. (*left & mid*): bigger models are more robust.

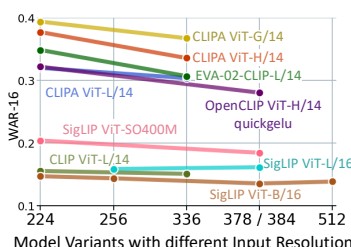 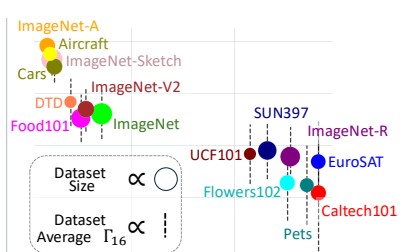 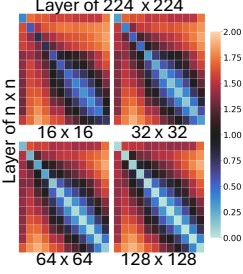

Figure 7: *Left*: **High Input Resolution Model** are less robust. *Mid*: **t-SNE of Dataset robustness** Dataset represented via 66 models robustness ($\Gamma_{16}^{D}$), indicates 3 clusters. *Right*: **Layers-wise features L2 similarity**: $n \times n$ model layers similarity w/ $224 \times 224$ ones, for EVA02-B-16. For a given heatmap (*e.g.* $16 \times 16$), the lower right indicates the similarity of deeper layers (brighter means more similar), while the upper left represents non-similar shallow layers (dull means less similar).

## 4 BENCHMARKING ANALYSIS

**Proposed WAR Metrics**: Spearman correlation between the rankings of 66 models, calculated using SAR and WAR averaging of relative robustness $\Gamma_{16}^{D}$ (across all datasets), and the individual dataset rankings is shown in fig. 5(left). WAR shows a slight decrease in avg. correlation (SAR-16 *0.89* vs WAR-16 *0.87*), but it also improves the representation of EuroSAT & ImageNet-A. The correlation score for EuroSAT increased from a weak/no correlation of 0.26 to a moderate 0.49.

**Model Architecture / Pretraining:** Figure 5 (right, (i)) shows, on average, **larger model** (x-axis) are **more robust**. Among the models, CLIP-ResNets (stars) are the least robust (compared to transformers (dots)) while EVA, MetaCLIP, CLIPA, and OpenCLIP exhibit the highest robustness against the LR. Higher GFLOP (size of dots) weakly impacts robustness with too many exceptions.

**Pretraining 'Quality over Quantity'**: Figure 5 (right (ii)) shows pre-training dataset size weakly correlates with robustness, with exceptions like SigLIP (10B), and M2-encoder (6B) performing worse. Models pre-trained on DataComp-1B generally outperform those pre-trained on LAION-2B, despite having over 500M fewer image-text pairs (fig. 6 (left)). This suggests that the **model** and **quality of pre-training** have a greater impact on robustness **than the quantity of pre-training**.

**Model Specific**: We remove architectural size advantages by categorizing top-performing models into parameter buckets as shown in fig. 6 (mid). For smaller models (150M and 430M parameters), OpenCLIP matches EVA and outperforms MetaCLIP and CLIPA, despite these two being built on top of OpenCLIP. However, for larger models, this trend reverses, with EVA-CLIP remaining superior for comparable sizes. Two factors contribute to performance discrepancies within models of the same parameter size: **(1) Fine-tuning:** ALBEF and BLIP fine-tuned variants are less robust on EuroSAT and Aircraft, reducing their overall robustness (fig. 6 (right)) **(2) Higher input resolution:** Models with higher input resolutions (*e.g.* $336 \times 336$) are generally less robust than their $224 \times 224$ counterparts, likely due to increased interpolation from $16 \times 16$ to higher resolutions (fig. 7 (left)).

**Dataset Specific**: Relative robustness of 66 models on each dataset forms its robustness vector representations. Representing these vectors using t-SNE (fig. 7 (mid)), reveal three major dataset clusters: high-robustness (long bars) (*e.g.* Caltech101), weakly robust (medium bars) (*e.g.* ImageNet), and least robust (smallest bar) (*e.g.* , ImageNet-A). This indicates that **low-resolution performance varies by dataset**, which warrants a deeper dive into dataset-specific robustness, *left as future work*.

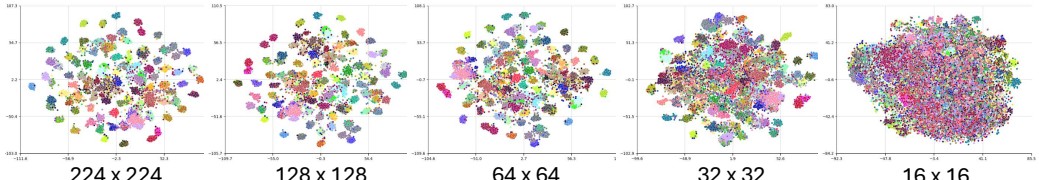

Figure 8: **Feats t-SNE**: EVA-02-CLIP-B/16 test features for Food-101, colored using class labels. With low resolutions (16×16, and 32×32), features become indistinguishable, thereby overlapping.

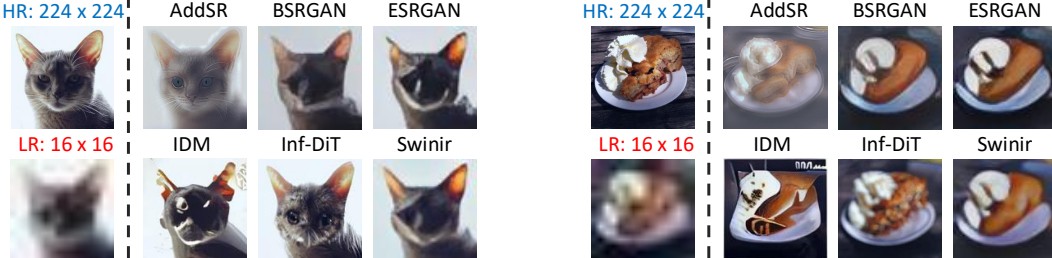

Figure 9: **Super resolution at 16×16**: Image from Pets (*left*) and Food102 (*right*). Models include AddSR [2024], BSRGAN [2021], ESRGAN [2018], IDM [2023], Inf-DiT [2024], and Swinir [2021].

**Inside Model**: Figure 1 shows the accuracy of all models first drops at 64×64, with a more significant decline after 32×32. EVA-B/16 features t-SNE (fig. 8) shows **features become indistinguishable as resolution decreases**. Inside the model, Figure 7 (right) shows the pairwise similarity (L2 distance) (Kornblith et al., 2019) between layers of models trained at different resolutions with the 224×224. Diagonal elements ($i^{th}$ layer of $n \times n$ model similarity with $i^{th}$ layer of a model trained at 224×224), is **more similar towards the deeper end** (lower right, the similarity is brighter), **than the initial layers** (upper left, the similarity is dull). Additionally, model similarity increases with resolution, while layers remain differentiable at all resolutions (dull non-diagonal values).

## 5 PROPOSED METHOD: LR-TK0

Figure 2 reveals two key insights: i) LR lacks fine-grained details ii) FM(s) make semantically reasonable predictions even at 16×16, highlighting the importance of preserving semantic capabilities (pre-training). While super-resolution (SR) methods could restore lost details without affecting models, zero-shot SR for very low resolutions ($\leq 64 \times 64$), doesn't work well in practice, as shown in fig. 9, where SR models fail to reconstruct out-of-domain images at 16×16. To enhance model robustness against low resolution, our solution **LR-TK0** adds trainable LR tokens on top of frozen transformers (preserving the pre-trained weights). These LR tokens learn to bridge the gap between the high-resolution (HR) and low-resolution (LR) domains, via self-supervised distillation (section 5.1). We train these tokens on synthetically generated diffusion-based images (Section 5.2) in a task-agnostic setting, ensuring the model is not exposed to any of the 15 target datasets.

### 5.1 LR TOKENS

To preserve the zero-shot capabilities of the model; *pre-trained weights of the model are frozen*. Instead, additional trainable tokens, referred to as "LR Tokens", are added on top of the spatial tokens after RGB to patch tokens conversion (patchification) and before each transformer block. As shown in fig. 10 (left) # LR tokens = # Spatial tokens × (N+1) blocks. These tokens aim to compensate for the loss of details in low resolution, thereby enhancing the model's interpretability of LR images. Contrary to prompt learning (Jia et al., 2022), where task-specific tokens are concatenated to the spatial tokens, ours are added/merged. Figure 7 (right) indicates LR feature at the initial layer deviates more than the later ones, thus LR tokens are added at every block.

**LR-TK0 Technique**: We adopt the multi-scale paradigm (Chen et al., 2019) *i.e.* training multiple low resolutions per HR image, given its success in the LR domain. Model without LR tokens (frozen pre-trained weights) as a teacher generating feature representations for HR images, as true embedding $f_{HR}^{\mathbf{T}}$. In contrast, LR tokens (& pre-trained model) act as student, generating

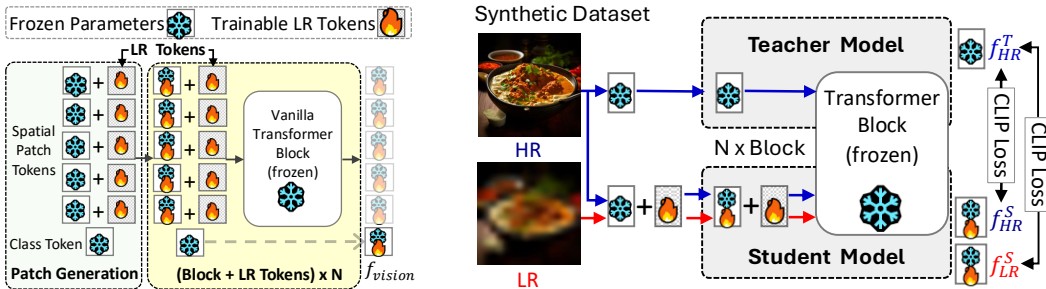

Figure 10: Fire (& ice) icons represent trainable (& frozen) parameters. *Left:* **LR tokens** are added to the frozen spatial patches (white) after patch generation, before each frozen transformer block, and class token as a final feature. *Right:* **LR-TK0:** Multi-scale training (only 1 shown for simplicity). Teacher (w/o LR tokens) generates $f_{HR}^T$ (HR), Student (w/ LR tokens) generates both $f_{HR}^S, f_{LR}^S$.

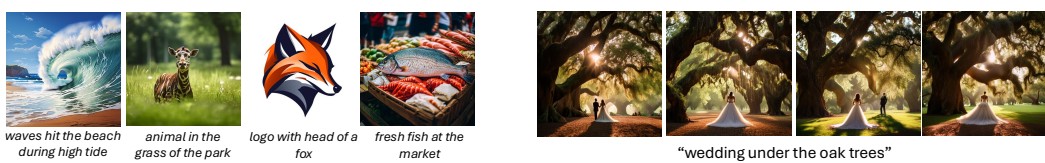

Figure 11: **Synthetic Images**: *(Left)* Images generated using PIXART-$\alpha$ [2023a] using randomly sampled captions from Conceptual Captions [2018]. *(Right)* Multiple images per caption.

embeddings for both HR ($f_{HR}^S$) and LR images ($f_{LR}^S$) as shown in fig. 10 (right). $f_{HR}^S$ & $f_{LR}^S$ are matched with $f_{HR}^T$ using a contrastive loss (Radford et al., 2021), similar to text and image alignment. Anchoring HR-LR features around frozen teacher avoids direct matching of HR-LR embeddings, preventing pulling the HR features towards LR ones (converging into one) (Khalid et al., 2020). This also ensures features w/ and w/o spatial tokens remain similar (regularization). Feature matching doesn't require any labels for these synthetic images, aka **unsupervised**. It also **task agnostic**, *i.e.* doesn't involve any model task-related characteristics (classification in this case).

## 5.2 SYNTHETIC HR DATASET

We use the diffusion model PIXART-$\alpha$ (Chen et al., 2023a) to generate synthetic HR images, via 7,000 randomly sampled captions from Conceptual Captions (Sharma et al., 2018). We expand our training set by creating multiple images (variations, human observation) per caption as shown in fig. 11. Conceptual Captions are commonly used in pretraining many zero-shot models (table 1), and using synthetic diffusion-based images helps LR tokens capture a wide range of domains, ensuring generalized training. **Random captions avoid targeting any specific dataset**. To our knowledge, our work is the first to train a model on synthetic diffusion images for zero-shot evaluation, contrary to training on a subset of target datasets (Chen et al., 2024). Following the multi-scale paradigm, we downsample HR images to a randomly sampled spatial resolution (height = width) from three LR resolution buckets [16,32], [32, 64], [64, 128], forming HR-LR image pairs.

**Zero-Shot:** If 7,000 (or fewer) concepts/captions can consistently enhance model performance across 15 datasets, it suggests that the model is likely learning the relationship between HR and LR features rather than exploiting shortcuts. This is supported by greater improvements at LR ($16\times16$) compared to HR ($128\times128$). If the model somehow cheats the zero-shot evaluation using diffusion-generated images, we would expect similar or better performance improvements at HRs.

## 6 PROPOSED METHOD: EXPERIMENTATION & ABLATION

**Implementation Details** Models are trained with 7K captions (& 30 images/captions) in a multi-scale paradigm. EVA is trained for 200 epochs, while MetaCLIP and OpenCLIP are for 10 epochs. Evaluation metrics (section 3): SAR (simple averaging of $\gamma_n^D$), WAR (weighted averaging of $\Gamma_n^D$), and Acc (average top-1). Higher number means better performance. Vanilla model's HR accuracy computes the accuracy gap $\mathcal{E}_D$, and dataset weights derived for $16\times16$ used for all resolutions (more in *Supplementary*). **'EVA-02-CLIP-B/16' (EVA-B/16)**, is used for all our model-level analysis.

Table 2: **LR-TK0 improvement on Foundation models:** 'Meta-B/16': MetaCLIP-ViT-B/16 (2.5B), 'OC-B/16': OpenCLIP-ViT-B/16. Higher number ∝ better performance.

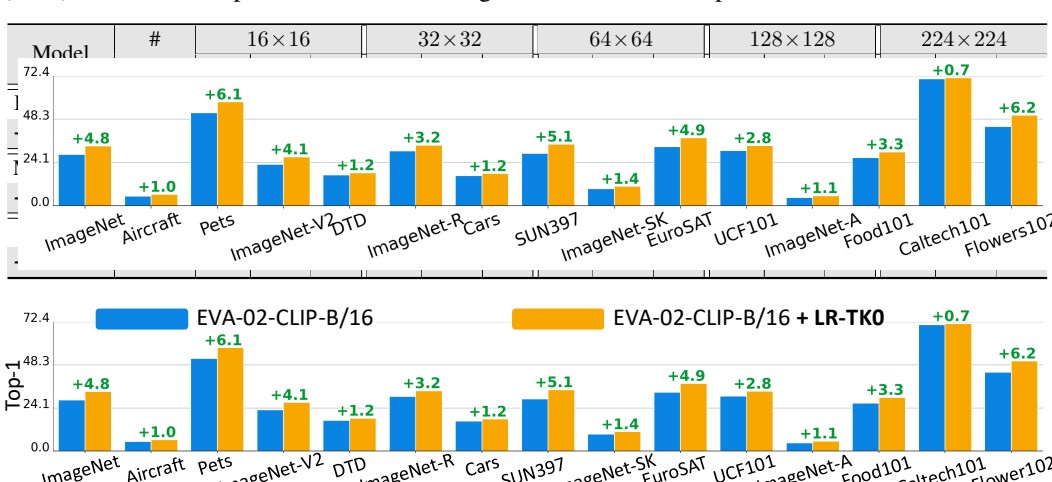

Figure 12: **Baseline vs LR-TK0**: Top-1 accuracy for EVA-B/16 on 16×16. (more in *Supplementary*)

Table 3: **Comparison with SR methods**: EVA-B/16 results, SR-specific pre-processing.

| Method | 16×16 | | | 32×32 | | |
|---|---|---|---|---|---|---|
| | SAR | WAR | Acc | SAR | WAR | Acc |
| Baseline | 34.1 | 26.8 | 25.0 | 71.8 | 59.0 | 51.2 |
| BSRGAN | 12.4 | 12.2 | 8.8 | 37.3 | 28.7 | 26.9 |
| ESRGAN | 14.2 | 15.1 | 10.0 | 40.3 | 32.6 | 28.9 |
| Swinir | 17.9 | 17.6 | 12.7 | 47.7 | 38.3 | 34.3 |
| AddSR | 20.5 | 16.8 | 15.0 | 48.3 | 36.0 | 35.2 |
| Inf-DiT | 29.0 | 25.3 | 20.9 | 67.7 | 58.6 | 48.0 |
| Our | **38.9** | **29.5** | **28.4** | **73.1** | **62.0** | **52.0** |

Table 4: **Generalization of LR-TK0 with other Zero-Shot Techniques**: Visual prompt Tuning (VPT) [2022] concatenates 50 learnable tokens to spatial tokens. RobustSAM [2024] is an image segmentation model modified for classification.

| LR-TK0 | WAR | | | | | SAR |
|---|---|---|---|---|---|---|
| | 16 | 32 | 64 | 128 | 224 | 16 |
| Baseline | 30.7 | 64.8 | 85.8 | 96.1 | 100 | 38.0 |
| +VPT | 35.5 | 64.1 | 84.6 | 94.5 | 97.8 | 42.6 |
| +RobustSAM | 32.2 | 61.5 | 82.7 | 92.4 | 93.0 | 37.8 |
| +LR Tokens | 35.4 | 66.4 | 85.9 | 95.5 | 98.7 | 42.4 |

## 6.1 RESULTS

**Table 2** shows our LR tokens consistently enhance robustness at low resolutions (16×16 & 32×32), particularly for MetaCLIP. While the low resolution is often seen as a domain shift problem (Ge et al., 2020), leading to potential declines in HR performance, our multi-scale training and HR teacher distillation minimize accuracy drops at higher resolutions (1-2% accuracy drop). Also, LR tokens have a minimal parameter gain (+3%). **Figure 12** shows Top-1 accuracy for EVA-B/16 with and without our LR-TK0, at 16×16, with max improvement on Flower-102 (6.2%). **Table 3** compares EVA-B/16 with super-resolution (SR) methods, with SR methods performing poorly in zero-shot settings for very low resolutions (fig. 9). In contrast, our approach is better suited for zero-shot scenarios. Diffusion-based SR method IDM is too computationally expensive to evaluate on large datasets like ImageNet (results in *Supplementary*). **Table 4** applies our LR-TK0 technique to visual prompt tuning which concatenates tokens (instead of adding) only before the first block. RobustSAM (segmentation models) modified for image classification (*Supplementary*).

## 6.2 ABLATION STUDY

**Design Choices: Table 5** shows not freezing the pre-trained weights (*i.e.* fine-tuning the last 4 blocks at 1/100 of the default learning rate) with and without LR tokens (first two rows) degrades the performance, indicating the necessity of preserving pre-trained weights. Our design choice is task agnostic *i.e.* model's classification plays no role in learning the HR-LR relationship but classifying LR images into captions (as class labels, *task-oriented*) has more or less the same performance. **Table 6** shows benefit of multi-scale training (3 buckets, faster to train).

Table 5: **Ablation:** EVA-B/16 trained with 7K captions and 50 images/caption. 'CL': use of classifier. *Not* frozen means fine-tuning end-to-end.

| Frozen | LR Tk. | CL | SAR-16 | WAR-16 | SAR-32 | WAR-32 |
|--------|--------|-----|--------|--------|--------|--------|
| Baseline (frozen) | | | 38.0 | 30.7 | 74.4 | 64.8 |
| | | | 31.1 | 24.5 | 67.2 | 56.6 |
| | ✓ | | 32.8 | 27.8 | 68.1 | 58.3 |
| ✓ | ✓ | | 42.3 | 35.2 | 75.3 | 66.4 |
| ✓ | ✓ | ✓ | 42.0 | 34.7 | 75.2 | 65.9 |

Table 6: **Multi-Scale (MS) Buckets**: '+' indicates Cumulative addition. E.g. [64,128] has [16,32] and [32,64] buckets.

| MS Buckets | WAR-16 | WAR-32 |
|------------|--------|--------|
| Baseline | 30.74 | 64.81 |
| [16, 32] | 34.01 | 64.77 |
| + [32, 64] | 35.28 | 66.10 |
| + [64, 128] | 35.45 | **66.40** |
| + [128, 224] | **35.73** | 65.91 |

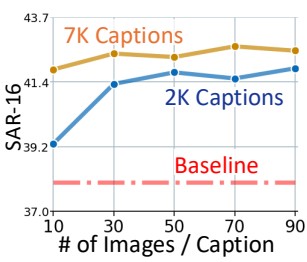

Figure 13: **#Images/Caption:** Robustness vs. Size of diffusion generated dataset.

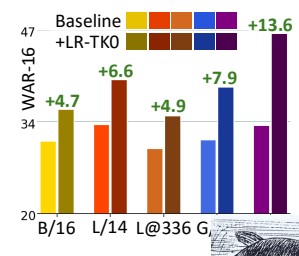

Figure 14: **LR-TK0 improves all EVA backbones**: L@336 is L/14 with 336 input

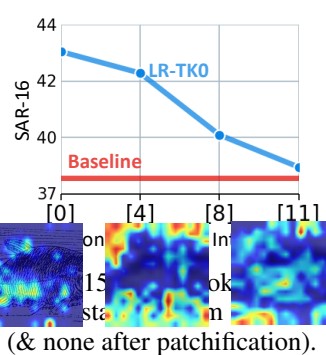

Figure 15: [$i$] LR tokens introduced starting from $i^{th}$ block (& none after patchification).

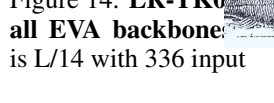
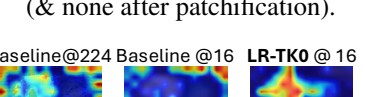
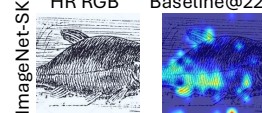
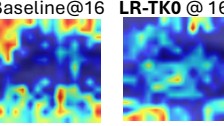
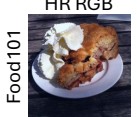
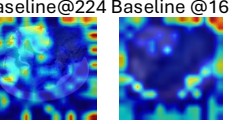
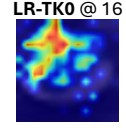

Figure 16: **LR token Grad-CAM**: Baseline (EVA-B/16) attention is scattered at $16\times16$ (compared to $224\times224$). LR-TK0 focuses on the object, likely capturing fine-grained details. @: input resolution.

**# Images/Caption: Figure 13** shows multiple images per caption & even 2000 captions consistently improve performance across 15 datasets, hinting at bridging the gap between HR-LR domains.

**EVA backbones: Figure 14** shows LR tokens enhance various EVA backbones, namely, Base (B/16), Large (L/14 & L@336), and G (G/14 & G/14+). Larger backbones, B<L<G, benefit from more tokens (via more layers). Model with $336\times336$ input underperforms (validation, fig. 7 (left)).

**Position of LR Tokens: Figure 15** shows introducing tokens in the earlier layer (starting from $[i]$-th block, and subsequent layers) is more helpful than later. This helps validate the observation in fig. 7 (right), *i.e.* initial layers suffer more at low resolution than deeper ones, validating the choice of fixing (introducing tokens) at initial layers than just at final features.

**Grad-CAM results:** On low resolutions of $16\times16$, vanilla model attention is dispersed and not as concentrated as $224\times224$ **(fig. 16)**. However, our method (w/ LR tokens) shows focus on the object which helps to learn better representations at low resolution.

## 7 CONCLUSION

Our extensive evaluation of Visual-Language Foundation Models through the LR0.FM benchmark has highlighted critical limitations in their ability to generalize under low-resolution conditions, a prevalent issue in real-world scenarios. While larger models and higher-quality pre-training datasets offer increased robustness, our findings underscore the significant impact of fine-tuning and input resolution on performance. Importantly, we observed that low-resolution inputs primarily disrupt the early layers of these models, leading to degraded performance. To address these challenges, we introduced the LR-TK0 strategy, which improves model robustness to low-resolution inputs without altering pre-trained weights, offering a practical solution for real-world applications. Additionally, our proposed Weighted Aggregated Robustness metric provides a more comprehensive evaluation of model resilience, addressing the limitations of existing metrics.

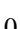

## 8    ACKNOWLEDGMENT

The authors thank Steven Dick (UCF High-Performance Computing) and Rohit Gupta (UCF CRCV) for their help in generating the synthetic data set.

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
