| Model | # Param | 16×16 | | | 32×32 | | | 64×64 | | | 128×128 | | | 224×224 | | |
|---|---|---|---|---|---|---|---|---|---|---|---|---|---|---|---|---|
| | | SAR | WAR | Acc | SAR | WAR | Acc | SAR | WAR | Acc | SAR | WAR | Acc | SAR | WAR | Acc |
| EVA-B/16 | 149.7M | 38.0 | 30.7 | 28.1 | 74.4 | 64.8 | 53.5 | 92.4 | 85.8 | 65.2 | 98.4 | 96.1 | 68.8 | 100 | 100 | 69.6 |
| **+LR-TK0** | 155.2M | 42.4 | 35.4 | 31.3 | 75.3 | 66.4 | 54.1 | 91.8 | 85.9 | 64.8 | 97.8 | 95.5 | 68.3 | 99.1 | 98.7 | 69.0 |
| Meta-B/16 | 149.6M | 32.1 | 27.2 | 23.4 | 65.3 | 54.4 | 47.0 | 89.5 | 83.6 | 62.9 | 98.5 | 96.7 | 68.5 | 100 | 100.0 | 69.4 |
| **+LR-TK0** | 151.6M | 41.9 | 38.9 | 30.2 | 71.7 | 66.0 | 51.0 | 89.3 | 85.4 | 62.6 | 96.7 | 95.4 | 67.3 | 97.6 | 97.4 | 67.9 |
| OC-B/16 | 149.6M | 33.4 | 26.5 | 24.8 | 68.6 | 59.5 | 49.8 | 89.2 | 84.1 | 63.6 | 96.8 | 94.8 | 68.3 | 100 | 100 | 70.4 |
| **+LR-TK0** | 151.6M | 37.4 | 34.4 | 27.4 | 69.0 | 63.0 | 49.9 | 88.8 | 84.2 | 63.4 | 96.8 | 95.1 | 68.4 | 99.0 | 99.0 | 69.8 |

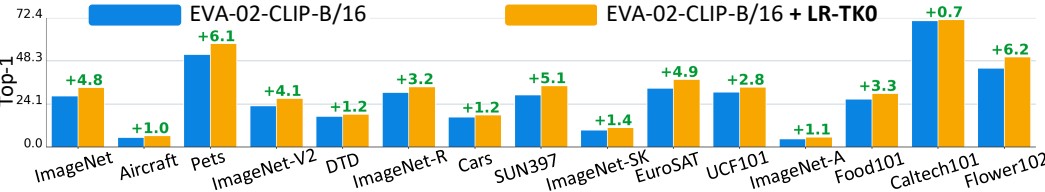

Figure 12: **Baseline vs LR-TK0**: Top-1 accuracy for EVA-B/16 on 16×16. (more in *Supplementary*)

Table 3: **Comparison with SR methods**: EVA-B/16 results, SR-specific pre-processing.

| Method | 16×16 | | | 32×32 | | |
|---|---|---|---|---|---|---|
| | SAR | WAR | Acc | SAR | WAR | Acc |
| Baseline | 34.1 | 26.8 | 25.0 | 71.8 | 59.0 | 51.2 |
| BSRGAN | 12.4 | 12.2 | 8.8 | 37.3 | 28.7 | 26.9 |
| ESRGAN | 14.2 | 15.1 | 10.0 | 40.3 | 32.6 | 28.9 |
| Swinir | 17.9 | 17.6 | 12.7 | 47.7 | 38.3 | 34.3 |
| AddSR | 20.5 | 16.8 | 15.0 | 48.3 | 36.0 | 35.2 |
| Inf-DiT | 29.0 | 25.3 | 20.9 | 67.7 | 58.6 | 48.0 |
| Our | **38.9** | **29.5** | **28.4** | **73.1** | **62.0** | **52.0** |

Table 4: **Generalization of LR-TK0 with other Zero-Shot Techniques**: Visual prompt Tuning (VPT) [2022] concatenates 50 learnable tokens to spatial tokens. RobustSAM [2024] is an image segmentation model modified for classification.

| LR-TK0 | WAR | | | | | SAR |
|---|---|---|---|---|---|---|
| | 16 | 32 | 64 | 128 | 224 | 16 |
| Baseline | 30.7 | 64.8 | 85.8 | 96.1 | 100 | 38.0 |
| +VPT | 35.5 | 64.1 | 84.6 | 94.5 | 97.8 | 42.6 |
| +RobustSAM | 32.2 | 61.5 | 82.7 | 92.4 | 93.0 | 37.8 |
| +LR Tokens | 35.4 | 66.4 | 85.9 | 95.5 | 98.7 | 42.4 |

## 6.1 RESULTS

**Table 2** shows our LR tokens consistently enhance robustness at low resolutions (16×16 & 32×32), particularly for MetaCLIP. While the low resolution is often seen as a domain shift problem (Ge et al., 2020), leading to potential declines in HR performance, our multi-scale training and HR teacher distillation minimize accuracy drops at higher resolutions (1-2% accuracy drop). Also, LR tokens have a minimal parameter gain (+3%). **Figure 12** shows Top-1 accuracy for EVA-B/16 with and without our LR-TK0, at 16×16, with max improvement on Flower-102 (6.2%). **Table 3** compares EVA-B/16 with super-resolution (SR) methods, with SR methods performing poorly in zero-shot settings for very low resolutions (fig. 9). In contrast, our approach is better suited for zero-shot scenarios. Diffusion-based SR method IDM is too computationally expensive to evaluate on large datasets like ImageNet (results in *Supplementary*). **Table 4** applies our LR-TK0 technique to visual prompt tuning which concatenates tokens (instead of adding) only before the first block. RobustSAM (segmentation models) modified for image classification (*Supplementary*).

## 6.2 ABLATION STUDY

**Design Choices: Table 5** shows not freezing the pre-trained weights (*i.e.* fine-tuning the last 4 blocks at 1/100 of the default learning rate) with and without LR tokens (first two rows) degrades the performance, indicating the necessity of preserving pre-trained weights. Our design choice is task agnostic *i.e.* model's classification plays no role in learning the HR-LR relationship but classifying LR images into captions (as class labels, *task-oriented*) has more or less the same performance. **Table 6** shows benefit of multi-scale training (3 buckets, faster to train).

Table 5: **Ablation:** EVA-B/16 trained with 7K captions and 50 images/caption. 'CL': use of classifier. *Not* frozen means fine-tuning end-to-end.

| Frozen | LR Tk. | CL | SAR-16 | WAR-16 | SAR-32 | WAR-32 |
|--------|--------|-----|--------|--------|--------|--------|
| Baseline (frozen) | | | 38.0 | 30.7 | 74.4 | 64.8 |
| | | | 31.1 | 24.5 | 67.2 | 56.6 |
| | ✓ | | 32.8 | 27.8 | 68.1 | 58.3 |
| ✓ | ✓ | | 42.3 | 35.2 | 75.3 | 66.4 |
| ✓ | ✓ | ✓ | 42.0 | 34.7 | 75.2 | 65.9 |

Table 6: **Multi-Scale (MS) Buckets**: '+' indicates Cumulative addition. E.g. [64,128] has [16,32] and [32,64] buckets.

| MS Buckets | WAR-16 | WAR-32 |
|------------|--------|--------|
| Baseline | 30.74 | 64.81 |
| [16, 32] | 34.01 | 64.77 |
| + [32, 64] | 35.28 | 66.10 |
| + [64, 128] | 35.45 | **66.40** |
| + [128, 224] | **35.73** | 65.91 |

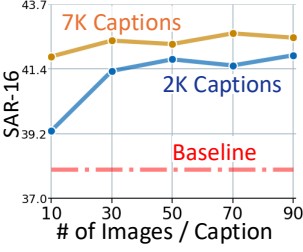

Figure 13: **#Images/Caption:** Robustness vs. Size of diffusion generated dataset.

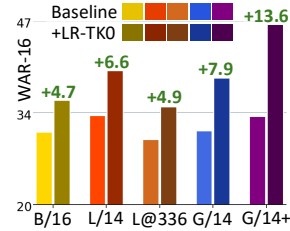

Figure 14: **LR-TK0 improves all EVA backbones**: L@336 is L/14 with 336 input

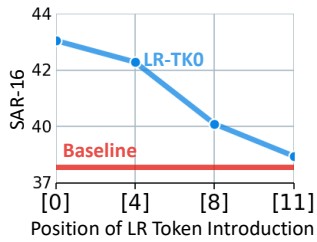

Figure 15: $[i]$ LR tokens introduced starting from $i^{th}$ block (& none after patchification).

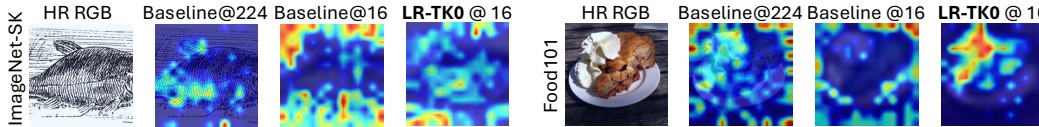

Figure 16: **LR token Grad-CAM**: Baseline (EVA-B/16) attention is scattered at $16 \times 16$ (compared to $224 \times 224$). LR-TK0 focuses on the object, likely capturing fine-grained details. @: input resolution.

**# Images/Caption: Figure 13** shows multiple images per caption & even 2000 captions consistently improve performance across 15 datasets, hinting at bridging the gap between HR-LR domains.

**EVA backbones: Figure 14** shows LR tokens enhance various EVA backbones, namely, Base (B/16), Large (L/14 & L@336), and G (G/14 & G/14+). Larger backbones, B<L<G, benefit from more tokens (via more layers). Model with $336 \times 336$ input underperforms (validation, fig. 7 (left)).

**Position of LR Tokens: Figure 15** shows introducing tokens in the earlier layer (starting from $[i]$-th block, and subsequent layers) is more helpful than later. This helps validate the observation in fig. 7 (right), *i.e.* initial layers suffer more at low resolution than deeper ones, validating the choice of fixing (introducing tokens) at initial layers than just at final features.

**Grad-CAM results:** On low resolutions of $16 \times 16$, vanilla model attention is dispersed and not as concentrated as $224 \times 224$ **(fig. 16)**. However, our method (w/ LR tokens) shows focus on the object which helps to learn better representations at low resolution.

## 7 CONCLUSION

Our extensive evaluation of Visual-Language Foundation Models through the LR0.FM benchmark has highlighted critical limitations in their ability to generalize under low-resolution conditions, a prevalent issue in real-world scenarios. While larger models and higher-quality pre-training datasets offer increased robustness, our findings underscore the significant impact of fine-tuning and input resolution on performance. Importantly, we observed that low-resolution inputs primarily disrupt the early layers of these models, leading to degraded performance. To address these challenges, we introduced the LR-TK0 strategy, which improves model robustness to low-resolution inputs without altering pre-trained weights, offering a practical solution for real-world applications. Additionally, our proposed Weighted Aggregated Robustness metric provides a more comprehensive evaluation of model resilience, addressing the limitations of existing metrics.

## 8    ACKNOWLEDGMENT

The authors thank Steven Dick (UCF High-Performance Computing) and Rohit Gupta (UCF CRCV) for their help in generating the synthetic data set.

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

# LR0.FM: Low-Res Benchmark and Improving robustness for Zero-Shot Classification in Foundation Models (Appendix)

## A  DATASET DESCRIPTION

This paper presents a comprehensive benchmarking of zero-shot image classification on low-resolution images utilizing 15 diverse datasets, each representing prominent computer vision challenges as depicted in Table 7. Among them, ImageNet Deng et al. (2009) stands out as a significant repository, containing 50,000 (in test-set) labeled images and serving as a standard for evaluating image classification models. Caltech101 Griffin et al. (2007), with its 6,085 test-set images spanning 101 object categories, is widely used for object recognition tasks. The Describable Textures Dataset (DTD) Cimpoi et al. (2014), comprising over 1,880 texture images in the test-set, facilitates texture analysis. Food101 provides 25,250 test-set images across 101 food categories, supporting food recognition tasks. SUN397's Zhou et al. (2014) 19,850 annotated test-set images aid scene recognition in understanding diverse environments. Stanford Cars Kramberger & Potočnik (2020) and FGVC Aircraft Maji et al. (2013) datasets focus on fine-grained classification tasks for vehicles and aircraft, respectively. Oxford Pets Parkhi et al. (2012) offers a dataset for pet breed classification, while Flower102 Liu et al. (2016) is dedicated to flower species recognition. Eurosat Helber et al. (2019) specializes in land use and cover classification using satellite imagery. UCF101 Soomro et al. (2012), containing over 1,794 video clips (in test-set), is pivotal for action recognition research, offering a diverse range of action sequences. Moreover, we explore four ImageNet variants for natural distribution shifts, previously considered as out-of-distribution (OOD) data for ImageNet Radford et al. (2021); Shu et al. (2022). ImageNet-V2 Recht et al. (2019) provides an independent test set with 10,000 natural images collected from different sources across 1,000 ImageNet categories, while ImageNet-A Hendrycks et al. (2021b) contains 7,500 challenging "natural adversarial examples" from 200 ImageNet categories misclassified by a standard ResNet-50 He et al. (2016). Lastly, ImageNet-R Hendrycks et al. (2021a) adds further diversity by offering 30,000 artistic renditions across 200 ImageNet categories, and ImageNet-Sketch Wang et al. (2019) includes 50,000 black-and-white sketches covering 1,000 categories, collected independently from the original ImageNet validation set. The test dataset size, the number of classes, and dataset focus are further elaborated in Table 7.

Table 7: **Statistics of benchmark datasets for zero-shot image recognition.**

| Dataset | Year | Test Size | # classes | Focus |
|---|---|---|---|---|
| ImageNet-A (2021b) | 2021 | 7500 | 200 | Generic |
| ImageNet-V2 (2019) | 2019 | 10,000 | 1000 | Generic |
| ImageNet (2009) | 2009 | 50,000 | 1000 | Generic |
| Caltech101 (2007) | 2004 | 6,085 | 101 | Generic |
| ImageNet-Sketch (2019) | 2019 | 50,000 | 1000 | Edges |
| ImageNet-R (2021a) | 2021 | 30,000 | 200 | Texture |
| EuroSAT (2019) | 2019 | 5,000 | 10 | Texture |
| DTD (2014) | 2014 | 1,880 | 47 | Edges, Texture |
| Food101 (2014) | 2014 | 25,250 | 101 | Fine-grained |
| Stanford Cars (2020) | 2013 | 8,041 | 196 | Fine-grained |
| FGVC-Aircraft (2013) | 2013 | 3,333 | 100 | Fine-grained |
| Oxford Pets (2012) | 2012 | 3,669 | 37 | Fine-grained |
| Oxford Flowers102 (2016) | 2008 | 6149 | 102 | Fine-grained |
| SUN397 (2014) | 2010 | 19,850 | 397 | Scene understanding |
| UCF101 (2012) | 2012 | 1,794 | 101 | Scene understanding |

**Table 8: Dataset templates:** As the main paper outlines, we adopt CLIP Radford et al. (2021) evaluation protocol for all models to ensure a fair comparison of low-resolution robustness. To generate the text embedding for a given image, we utilize dataset-specific templates, such as "a photo of a [label]", "a low-resolution photo of a [label]", *etc* as detailed in Table 8. For each class label, we generate multiple text embeddings by inserting the label into $n$ prompt templates and then

Table 8: **Benchmark Datasets Templates** Zero-shot image classification. Here **[L]** is the class name (labels). These templates are taken from CLIP (Radford et al., 2021) and OPENCLIP (Ilharco et al., 2021)

| Dataset | Sample prompt template | # Prompts |
|---|---|---|
| ImageNet | a low resolution photo of a **[L]**, a photo of a small **[L]**, art of a **[L]**, *etc.* | 80 |
| ImageNet-SK | a sketch of the **[L]**, a rendering of a **[L]**, a drawing of a **[L]**, *etc.* | 80 |
| ImageNet-A | a sculpture of a **[L]**, a close-up photo of the **[L]**, the cartoon **[L]** *etc.* | 80 |
| ImageNet-V2 | a black and white photo of a **[L]**, a **[L]** in a video game, a toy **[L]**, *etc.* | 80 |
| ImageNet-R | a cropped photo of the **[L]**, a blurry photo of the **[L]**, graffiti of a **[L]**, *etc.* | 80 |
| Caltech101 | a photo of a **[L]**, a painting of a **[L]**, the origami **[L]**, the toy **[L]**, *etc.* | 34 |
| DTD | a photo of a **[L]** texture, a photo of a **[L]** pattern, *etc.* | 8 |
| Food101 | a photo of **[L]**, a type of food | 1 |
| SUN397 | a photo of a **[L]**, a photo of the **[L]** | 2 |
| Cars | a photo of a **[L]**, a photo of my new **[L]**, a photo of my dirty **[L]**, *etc.* | 8 |
| Aircraft | a photo of a **[L]**, a type of aircraft & a photo of the **[L]**, a type of aircraft | 2 |
| Pets | a photo of a **[L]**, a type of pet | 1 |
| Flowers102 | a photo of a **[L]**, a type of flower | 1 |
| EuroSAT | a centered satellite photo of the **[L]**, a centered satellite photo of a **[L]**, *etc.* | 3 |
| UCF101 | a video of a person doing **[L]**, a example of a person practicing **[L]**, *etc.* | 48 |

average these $n$ embeddings. For instance, consider an image of a cat from the Imagenet dataset. With 1000 class labels and 80 prompt templates, we insert the label "cat" into the templates, generate 80 corresponding text embeddings, and compute their average to represent the cat class in text space. This process yields 1000 text embeddings, one for each class. The dot product between the image embedding and these 1000 text embeddings produces class logits, where the highest logit score determines the predicted class. In Table 8, we present data-specific prompt template samples along with the total number of such prompts.

## B  PERFORMANCE DROP

**Figure 17: Zero-shot Classification vs Resolution:** This figure is an extension of Figure 1 in the main paper, highlighting a major objective of our study: the relationship between resolution and model performance. As the resolution decreases, we observe a pronounced decline in the performance of all foundational vision-language models when compared to their high-resolution counterparts ($224 \times 224$), as illustrated in Figure 17. Our analysis reveals that this performance drop is consistent across 15 widely used computer vision benchmark datasets, affecting all model backbones. Notably, a performance decline begins at a resolution of $64 \times 64$, with a more substantial degradation occurring as the resolution falls below $32 \times 32$.

## C  WEIGHTED AGGREGATED ROBUSTNESS (WAR)

The improved relative robustness is computed as:

$$\Gamma_n^D = \gamma_n^D \times (1 - e^{-\alpha(\mathcal{E}_D)^2}) \quad | \quad \alpha >> 1 \quad \& \quad 0 \leq \mathcal{E}_D \leq 1 \tag{3}$$

The additional factor $(1 - e^{-\alpha\mathcal{E}_D^2})$ is shown in Figure 4 in the main paper for $\alpha = 200$. It remains close to 1 for the majority values of x or $\mathcal{E}_D$ but steeps to 0 as $\mathcal{E}_D$ reaches 0. In accuracy terms, as $A_{224}$ comes closer to $A_{rand}$, relative robustness starts dropping to 0. This is shown in Table 9 where the normal relative robustness score is high $\sim 20 - 40\%$ for almost random predictions, given the highest resolution accuracy is also close to random prediction. Our weighing term bring these scores to approximately $< 12\%$.

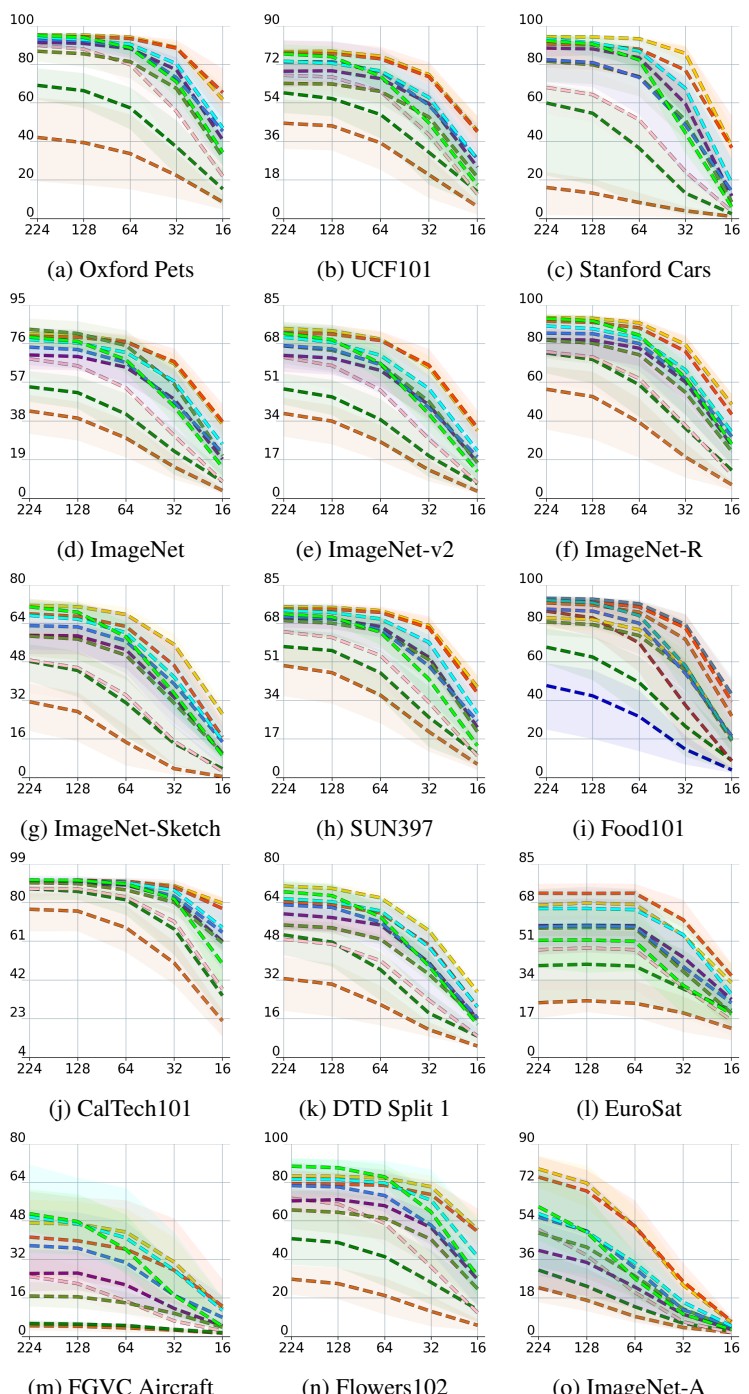

Figure 17: **Top-1 Accuracy drop**: Drop in accuracy for all models for all the datasets. The color scheme same as Figure 1 from the main submission.

Table 9: **Abnormally high relative robustness for random predictions**. Our All numbers in percentage (100%). We have shown results only for Easiness $\mathcal{E}_D < 0.15$, *i.e.* highest resolution accuracy ($A_{224}$) is close to random predictions. Our $\hat{\gamma}^n_{R,D}$ is plotted for $\alpha = 200$, *i.e.* $\hat{\gamma}^n_{R,D} = \gamma^n_{R,D} \times (1 - e^{-200\mathcal{E}_D^2})$. $A_{\text{rand}} = \frac{1}{\text{\# of classes}}$. High robustness scores within $2\times A_{\text{rand}}$ are bold. Lines are drawn for easy readability.

| Model | Dataset | $A_{\text{rand}}$ | $A_{224}$ | $A_{16}$ | $\gamma^{16}_{R,D}$ | $\Gamma^D_{16}$ (*Our*) | $A_{32}$ | $\gamma^{32}_{R,D}$ | $\Gamma^D_{32}$ (*Our*) | $A_{64}$ | $\gamma^{64}_{R,D}$ | $\Gamma^D_{64}$ (*Our*) | $A_{128}$ | $\gamma^{128}_{R,D}$ | $\Gamma^D_{128}$ (*Our*) |
|---|---|---|---|---|---|---|---|---|---|---|---|---|---|---|---|
| ALBEF (4M) | Cars | 0.5 | 2.0 | 0.6 | **28.8** | 1.2 | 1.0 | **51.5** | 2.2 | 1.3 | 66.2 | 2.8 | 1.6 | 83.3 | 3.5 |
| ALBEF (14M+flickr_finetuned) | Aircraft | 1.0 | 5.7 | 1.8 | **31.7** | 11.3 | 3.5 | 60.8 | 21.7 | 4.0 | 70.9 | 25.2 | 5.2 | 91.5 | 32.6 |
| BLIP-ViT-B/16 (129M + COCO) | Aircraft | 1.0 | 5.3 | 1.4 | **26.6** | 8.3 | 3.4 | 63.3 | 19.8 | 5.0 | 94.9 | 29.7 | 5.1 | 95.5 | 29.9 |
| ALBEF (14M) | Aircraft | 1.0 | 3.6 | 1.2 | **33.6** | 4.2 | 2.2 | 61.3 | 7.7 | 3.1 | 86.6 | 10.9 | 3.8 | 105.9 | 13.3 |
| ALBEF (4M) | Aircraft | 1.0 | 2.7 | 1.0 | **37.1** | 2.1 | 1.4 | **50.6** | 2.8 | 1.8 | **66.3** | 3.7 | 2.3 | 87.6 | 4.9 |
| BLIP-ViT-B/16 (129M) | Aircraft | 1.0 | 3.8 | 1.2 | **31.7** | 4.6 | 1.9 | **51.6** | 7.5 | 3.8 | 100.0 | 14.5 | 4.3 | 115.1 | 16.7 |
| ALBEF (14M + coco_finetuned) | Aircraft | 1.0 | 6.1 | 1.6 | **27.1** | 11.0 | 3.8 | 61.6 | 25.0 | 5.5 | 90.1 | 36.7 | 5.8 | 95.1 | 38.7 |
| BLIP-ViT-B/16 (4M) | Aircraft | 1.0 | 6.6 | 1.6 | **23.6** | 11.1 | 2.9 | 43.2 | 20.2 | 4.8 | 72.7 | 34.1 | 6.0 | 90.5 | 42.4 |
| BLIP-ViT-L/16 (129M + Flickr) | Aircraft | 1.0 | 5.9 | 1.9 | **31.8** | 12.4 | 2.8 | 47.5 | 18.4 | 4.7 | 79.3 | 30.8 | 4.9 | 82.3 | 32.0 |
| BLIP-ViT-B/16 & CapFilt-L (129M) | Aircraft | 1.0 | 5.0 | 1.4 | **27.5** | 7.6 | 2.6 | 52.1 | 14.4 | 3.8 | 75.4 | 20.9 | 4.3 | 86.2 | 23.9 |
| BLIP-ViT-B/16 (129M + Flickr) | Aircraft | 1.0 | 4.8 | 1.8 | **36.9** | 9.3 | 3.7 | 76.2 | 19.3 | 4.9 | 101.3 | 25.6 | 5.0 | 103.8 | 26.3 |
| BLIP-ViT-L/16 (129M) | Aircraft | 1.0 | 5.3 | 2.0 | **37.6** | 11.9 | 3.4 | 64.0 | 20.3 | 5.0 | 94.4 | 29.8 | 5.8 | 107.9 | 34.1 |
| ALBEF (14M) | EuroSAT | 10.0 | 17.4 | 17.3 | **99.1** | 66.3 | 15.7 | **89.7** | 60.1 | 17.8 | **102.1** | 68.4 | 20.7 | **118.8** | 79.5 |
| ALBEF (4M) | EuroSAT | 10.0 | 19.4 | 7.7 | **39.8** | 32.9 | 11.3 | **58.6** | 48.5 | 19.0 | **97.9** | 81.0 | 20.0 | **103.5** | 85.6 |

# D  CNN VS VIT

**Table 10:** While early research in multi-modal learning employed both CNN and ViT-based backbones (such as CLIP Radford et al. (2021) and OpenCLIP Ilharco et al. (2021)) – new SOTA models solely leverage ViTs as their backbone. We explore the effectiveness of CNN (mainly ResNets-based) and ViTs-based backbone within the same model settings while low-resolution shift occurs. Here, we found that ViT-based backbones (such as ViT-B/32, ViT-B/16, and ViT-L/14) are much more robust and lower sensitive to LR shift as compared to CNN-based (such as RN50, RN101, RN50x4, RN50x16, and RN50x64) backbones. In Table 10, we report the SAR and WAR ($\Gamma_n^D$) scores of CLIP Radford et al. (2021) backbones across 15 datasets for different severity labels.

Table 10: **Robustness analysis of CNN vs ViT-based backbones** of CLIP model across 15 datasets for different severity labels using $\Gamma_n^D(\uparrow)$.

| Backbones | # Params ($\downarrow$) | $A_{224}$ WAR | $224 \to 128$ SAR | WAR | $224 \to 64$ SAR | WAR | $224 \to 32$ SAR | WAR | $224 \to 16$ SAR | WAR | Avg. ($\uparrow$) SAR | WAR |
|---|---|---|---|---|---|---|---|---|---|---|---|---|
| RN50 | 102M | 99.90 | 92.54 | 87.89 | 70.16 | 66.00 | 32.75 | 32.52 | 10.00 | 17.20 | 51.36 | 50.90 |
| RN101 | 120M | 99.95 | 94.78 | 92.09 | 75.66 | 70.99 | 39.18 | 38.11 | 10.49 | 13.99 | **55.03** | **53.80** |
| RN50x4 | 178M | 99.99 | 92.46 | 88.82 | 70.94 | 66.50 | 34.77 | 30.64 | 10.04 | 11.88 | 52.05 | 49.46 |
| RN50x16 | 291M | 100 | 91.41 | 85.08 | 73.09 | 64.42 | 37.72 | 32.06 | 10.90 | 14.46 | 53.28 | 48.76 |
| RN50x64 | 623M | 100 | 93.58 | 88.49 | 78.70 | 70.26 | 44.22 | 35.11 | 12.09 | 14.11 | 57.15 | 52.24 |
| ViT-B/16 | 150M | 100 | 96.35 | 93.93 | 83.39 | 77.89 | 53.03 | 44.89 | 21.01 | 19.90 | 63.45 | 59.15 |
| ViT-B/32 | 151M | 99.98 | 96.62 | 94.88 | 82.67 | 77.68 | 52.39 | 44.49 | 19.41 | 16.91 | 62.77 | 58.49 |
| ViT-L/14 | 428M | 100 | 97.12 | 95.36 | 87.05 | 80.35 | 63.40 | 51.38 | 25.68 | 18.20 | **68.31** | **61.32** |
| ViT-L/14@336px | 428M | 100 | 96.08 | 93.74 | 85.68 | 78.22 | 61.11 | 50.65 | 24.42 | 17.81 | 66.82 | 60.10 |

# E  IMPLEMENTATION DETAILS

**Dataset Weights:** In Table 11, we have shown the dataset-specific weight values used to compute weighted aggregated robustness for low-resolution. All models were trained on 2 48GB GPUs.

Table 11: Optimized dataset weight values for WAR-16, shown using pie chart in Figure 4 (right) in the main paper.

| Dataset | Weight | SAR-16 Correlation | WAR-16 Correlation |
|---|---|---|---|
| Imagenet | 0.15556157429688613 | 0.99269 | 0.93295 |
| ImageNet-A | 0.970498446080589 | 0.55646 | **0.68070** |
| ImageNet-V2 | 0.2854574367981364 | 0.99165 | 0.93733 |
| ImageNet-R | 0.01 | 0.98201 | 0.90682 |
| ImageNet-Sketch | 0.021456095637452655 | 0.95086 | 0.87241 |
| Caltech101 | 0.01 | 0.97695 | 0.90853 |
| DTD split-1 | 0.505922498560715 | 0.87676 | 0.82507 |
| Food101 | 0.01 | 0.97771 | 0.91575 |
| SUN397 | 0.407563119725743 | 0.98760 | 0.94531 |
| Stanford Cars | 0.13583821249199218 | 0.96639 | 0.91721 |
| FGVC Aircraft | 0.8229545014750042 | 0.89746 | 0.89016 |
| Oxford Pets | 0.08995285864599148 | 0.97224 | 0.90114 |
| Flowers102 | 0.08972060770047119 | 0.97073 | 0.91809 |
| EuroSAT | 1.0 | 0.25753 | **0.49229** |
| UCF101 | 0.01 | 0.97324 | 0.93516 |

**Super Resolution Method Preprocessing:** Here, we present preprocessing steps for two pipelines *i.e.* (i) **Vanilla Pipeline:** raw image $\to$ create a low-resolution image using `transforms.Resize`$(\cdot) \to$ upscale it to the model resolution using `transforms.Resize`$(\cdot) \to$ input to the model; and (ii) **Super Resolution Pipeline:** raw image $\to$ create a low-resolution image using `transform_test`$(\cdot) \to$ pass through Super

Resolution models → get the model resolution using sr_transform_test(·) → input to the model. The detailed implementation of these two pipelines is illustrated in the code below:

Listing 1: SR data preprocessing

```python
# org_res is the original model resolution
# low_res is the low resolution
# normalize (mean, std) is the normalization specific to the model

# Pipeline-1: Vanilla
transform_test = transforms.Compose([
    transforms.Resize(low_res,interpolation=InterpolationMode.BICUBIC),
    transforms.Resize(org_res,interpolation=InterpolationMode.BICUBIC),
    transforms.CenterCrop(size=(org_resolution, org_resolution)),
    _convert_image_to_rgb, # converts img to RBG using PIL
    transforms.ToTensor(),
    normalize,
])

EVA_INPUT = transform_test(RGB_IMG)
...
# Pipeline-2: SUPER RESOLUTION
# RAW Image --> SR model
transform_test = transforms.Compose([
    transforms.Resize(low_res,interpolation=InterpolationMode.BICUBIC),
    transforms.CenterCrop(size=(low_res, low_res)),
    _convert_image_to_rgb,
    transforms.ToTensor(),
    normalize,
])

# SR Image --> EVA model
mean = (0.48145466, 0.4578275, 0.40821073)
std = (0.26862954, 0.26130258, 0.27577711)
sr_transform_test = transforms.Compose([
    transforms.Resize(org_res,interpolation=InterpolationMode.BICUBIC),
    transforms.CenterCrop(size=(org_res, org_res)),
    _convert_image_to_rgb,
    transforms.ToTensor(),
    transforms.Normalize(mean, std),
])

SR_INPUT = transform_test(RGB_IMG)
SR = SR_MODEL (SR_INPUT)
SR = transforms.functional.to_pil_image(normalize(SR), mode=None)
EVA_INPUT = sr_transform_test(SR)
....
```

**RobustSAM implementation for Classification:** We use the official code[3] to replace the mask token with the vision class token. Robust SAM is a segmentation model. We remove all its segmentation mask components and mask prediction step. The vision transformer encoder's last block is used instead of the decoder, and all the mask component is stripped away. Vanilla Transformer is treated as a teacher. In the student model, the class token is replaced with a *learnable token*. This new learnable token is passed through each transformer block. After the first block, we treat this as "early_feature" as mentioned in the official github. Using RobustSAM denoising trainable modules, we generate 'complementary_features' of these early features. After the final block, we use the new learnable token to generate 'final_image_embeddings' using the 'self.fourier_last_layer_features (image_embeddings, clear=CLEAR)'.

'robust_features = complementary_features + final_image_embeddings'.

MSE makes noisy and clear class token and robust features similar.

---

[3]URL: https://robustsam.github.io/

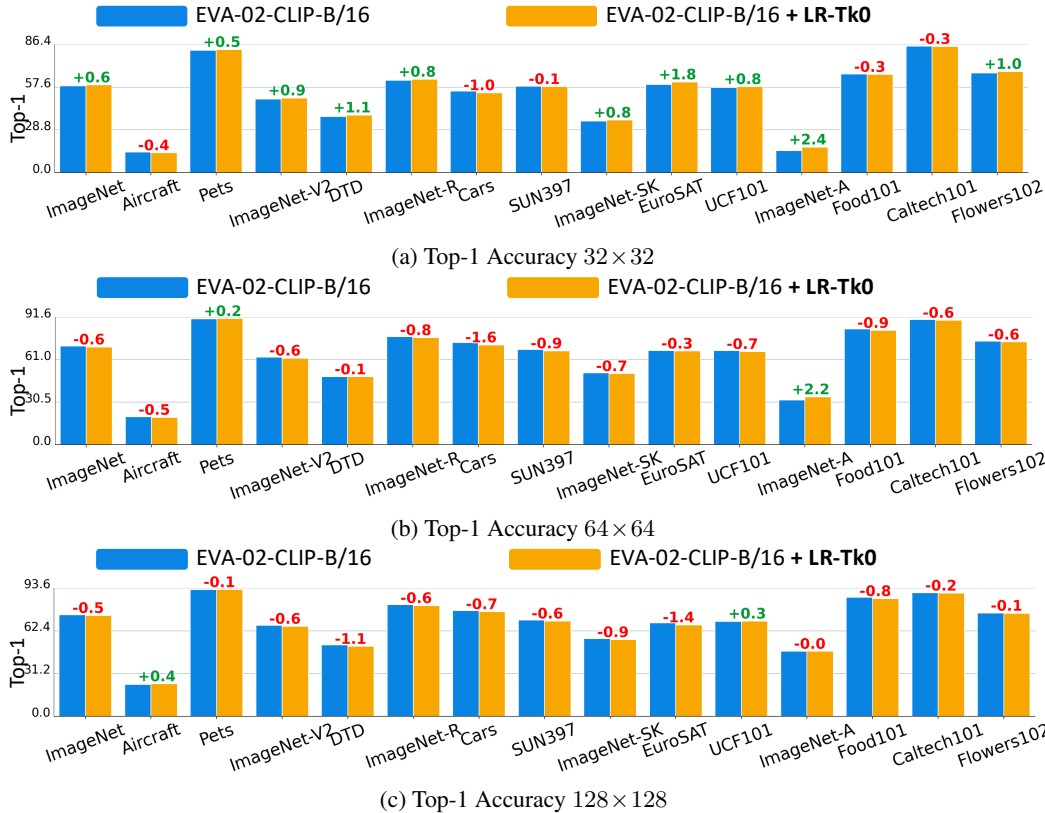

(a) Top-1 Accuracy $32 \times 32$

(b) Top-1 Accuracy $64 \times 64$

(c) Top-1 Accuracy $128 \times 128$

Figure 18: **Vanilla vs LR-TK0 (Our)**: Top-1 accuracy for `EVA-02-CLIP-B/16` model for different resolutions.

**VPT Implementation:** VPT is the same as ours, instead of adding on top of spatial tokens, trainable 50 tokens are concatenated to frozen spatial tokens before the first block. The decline in the performance at higher resolution indicates the need for introducing tokens at every layer instead of just once at the start.

Both methods follow the same training environment as our LR-TK0 (multi-training paradigm and diffusion-based images 7k * 30).

# F  MORE RESULTS

## F.1  DATASET WISE RESOLUTION VS. ACCURACY

In **Figure 18**, we highlight the superior zero-shot low-resolution performance (*i.e.* accuracy) of our proposed method, **LR-TK0**, compared to the vanilla `EVA-02-CLIP-B/16` model, while utilizing the same backbone across 15 datasets at varying resolutions: $32 \times 32$, $64 \times 64$, and $128 \times 128$. The main paper already demonstrates the results for the $16 \times 16$ resolution in Figure 12.

Since EVA performs far superior to random prediction, we present a detailed dataset-specific breakdown of gamma robustness, denoted as $\Gamma_n^D \approx \gamma_n^D$ for our proposed method compared with the vanilla `EVA-02-CLIP-B/16` across resolutions $n = 16, 32, 64$, and 128. These results are detailed in **Figure 19**. It should be noted that robustness is the absolute value and in Figure 19, robustness exceeds 100 only when the model's accuracy at lower resolutions surpasses its accuracy at the original 224 resolution.

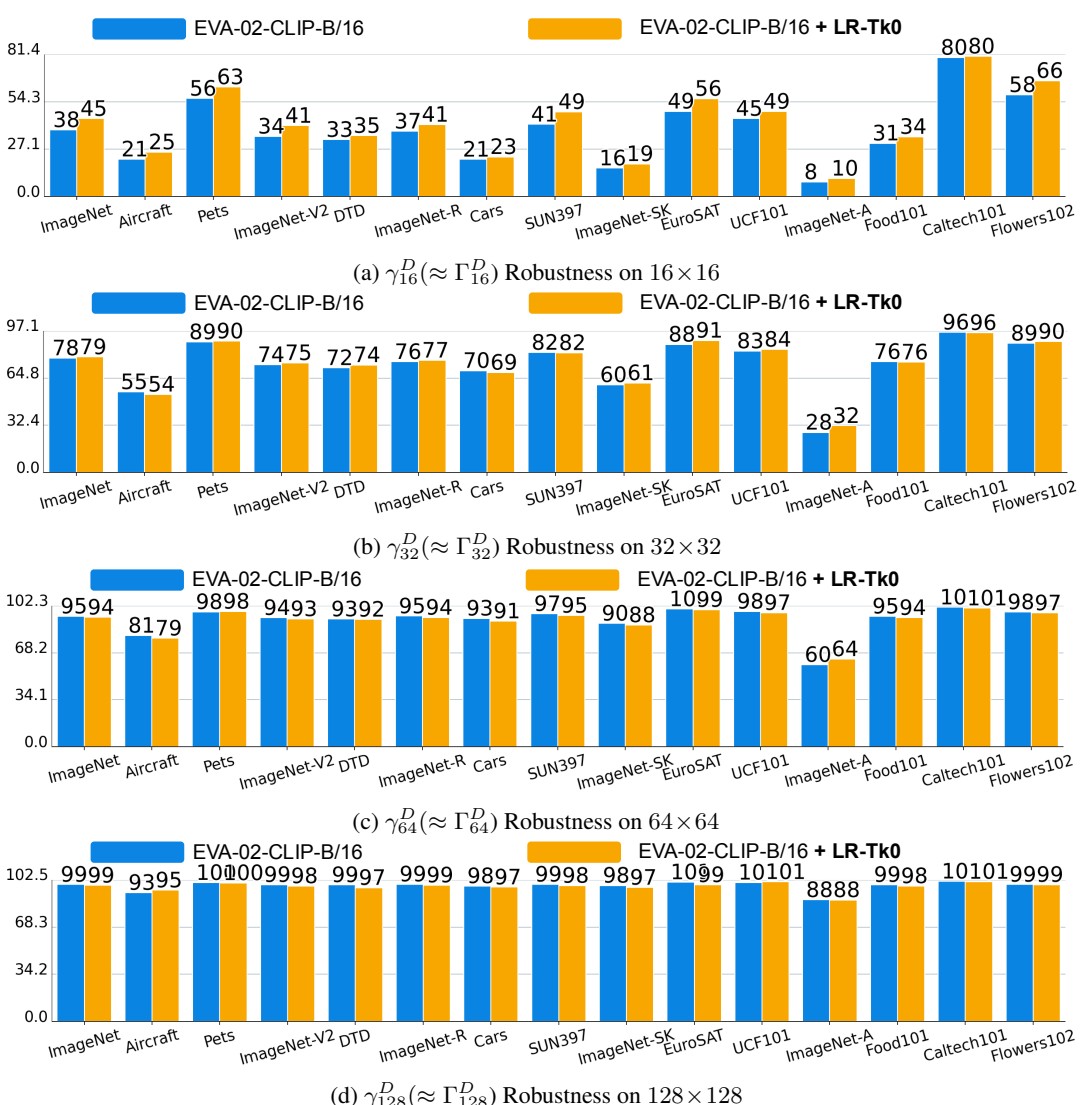

Figure 19: **Vanilla vs LR-TK0 (Our)**: Gamma Robustness for `EVA-02-CLIP-B/16` model for different resolutions on each dataset.

Table 12: **Comparison with SR**: EVA-B/16 results, with different data preprocessing (for SR).

| Method | 16×16 | | | 32×32 | | | 64×64 | | | 128×128 | | |
|---|---|---|---|---|---|---|---|---|---|---|---|---|
| | SAR | WAR | Acc | SAR | WAR | Acc | SAR | WAR | Acc | SAR | WAR | Acc |
| Baseline | 34.1 | 26.8 | 25.0 | 71.8 | 59.0 | 51.2 | **91.6** | 83.8 | **63.8** | **98.2** | **95.4** | **67.6** |
| BSRGAN [2021] | 12.4 | 12.2 | 8.8 | 37.3 | 28.7 | 26.9 | 70.1 | 58.0 | 49.4 | 88.9 | 77.2 | 61.9 |
| ESRGAN [2018] | 14.2 | 15.1 | 10.0 | 40.3 | 32.6 | 28.9 | 74.4 | 61.8 | 52.4 | 90.8 | 79.7 | 63.2 |
| Swinir [2021] | 17.9 | 17.6 | 12.7 | 47.7 | 38.3 | 34.3 | 79.2 | 68.9 | 55.6 | 92.7 | 84.6 | 64.2 |
| AddSR [2024] | 20.5 | 16.8 | 15.0 | 48.3 | 36.0 | 35.2 | 73.5 | 57.5 | 52.3 | 83.6 | 69.4 | 58.7 |
| Our | **38.9** | **29.5** | **28.4** | **73.1** | **62.0** | **52.0** | 91.4 | **85.5** | 63.6 | 97.6 | 95.2 | 67.3 |

## F.2 ALL SR RESULTS

In Table 12, we present a comparison of our proposed **LR-TK0** method against the baseline and several state-of-the-art super-resolution methods, including BSRGAN Zhang et al. (2021), ESR-

Table 13: IDM & Inf-DiT performance on Pets dataset.

| Method | Top -1 16×16 | Top -5 16×16 | Top-1 32×32 | Top-5 32×32 |
|---|---|---|---|---|
| Eva-B/16 | 51.840 | 84.710 | 82.530 | 98.530 |
| Eva-B/16 + LR-Tk0 | 57.92 | 88.66 | 83.07 | 98.36 |
| IDM + Eva-B/16 | 7.2 | 29.03 | 7.88 | 30.25 |
| Inf-DiT + Eva-B/16 | 29 | 60.94 | 73.43 | 94.36 |

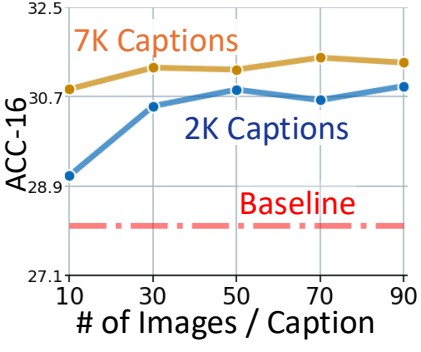 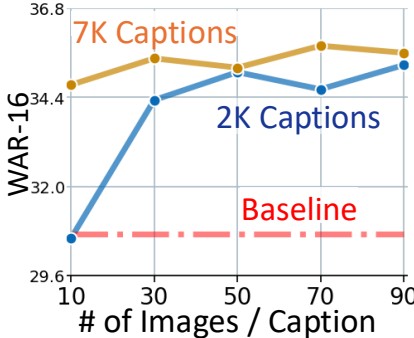

Figure 20: **Images/ Caption** : For ACC, and WAR, evaluation metrics on 16×16. SAR in the main paper.

GAN Wang et al. (2018), SwinIR Liang et al. (2021), and AddSR Xie et al. (2024). All super-resolution methods were employed in a zero-shot setting to ensure a fair comparison. Our method significantly outperformed these super-resolution techniques across all resolutions and demonstrated a substantial improvement over the baseline at resolutions of $16 \times 16$ and $32 \times 32$. Furthermore, it exhibited comparable robustness at resolutions of $64 \times 64$ and $128 \times 128$ with the baseline method.

## F.3 SR RESULTS FOR IDM AND INF-DIT

**Table 13**: IDM generalized Zero shot weights do not match their GitHub implementation. Hence we use their weight for cat datasets. We evaluate IDM on the pets dataset which is the closest to its pretrained weights. For uniformity, we compare Inf-DiT on the pets dataset as well. Both diffusion-based models take around 4-5 mins per batch of 10 images, making large-scale dataset evaluation impossible.

## F.4 GRAD CAM RESULTS

**Figure 23**, an extension of Figure 16 in the main paper, presents the Grad CAM visualization of the vanilla model and proposed method, showcasing the effect of proposed LR tokens.

# G ABLATIONS

**Number of Images Per Caption:** In the main paper, Figure 13 presents the number of generated (by diffusion model) images (captions) with SAR-16 metric to emphasize how it helps to improve the model robustness. Here, in **Figure 20**, we extend this by including ACC-16 and WAR-16 evaluation metrics, while varying the number of generated images.

**Hyperparameter** $\alpha$ signifies the rate of robustness declines as accuracy approaches random prediction. In **Figure 21**, we varied the $\alpha$ value with robustness and considered $\alpha = 200$ for our experiments as shown in Figure 4 (left) of the main paper.

**LR token position:** In the main paper, Figure 15 shows the performance (in terms of SAR-16, 16 is for resolution) with respect to the position of LR tokens being introduced in the form of a line chart. Here, in **Table 14**, we detailed the corresponding numerical values of Figure 15 for better clarity.

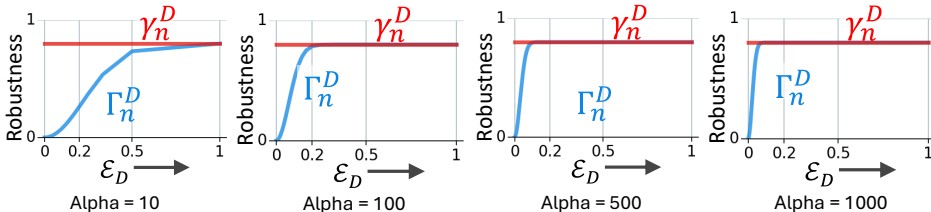

Figure 21: **Rate of robustness declines as accuracy approaches random prediction.**

Table 14: **LR token Position (Pos):** $[i]$ means LR tokens after $i^{th}$ block (and no token after patchification).

| Pos. | SAR-16 | WAR-16 | SAR-32 | WAR-32 |
|---|---|---|---|---|
| [0] | **42.4** | **35.4** | 75.3 | 66.4 |
| [5] | 41.4 | 35.3 | **75.4** | **67.0** |
| [8] | 39.6 | 33.3 | 74.8 | 65.5 |
| [11] | 38.4 | 31.3 | 74.5 | 64.6 |

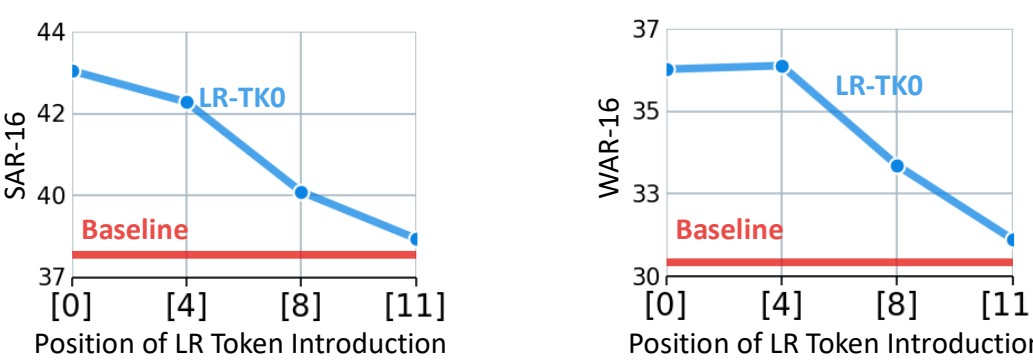

Figure 22: **Position of LR tokens introduction**: No tokens were added after the position embedding stage. [i]-th indicates the block from which LR tokens were introduced. Performance metrics variants of Figure 15.

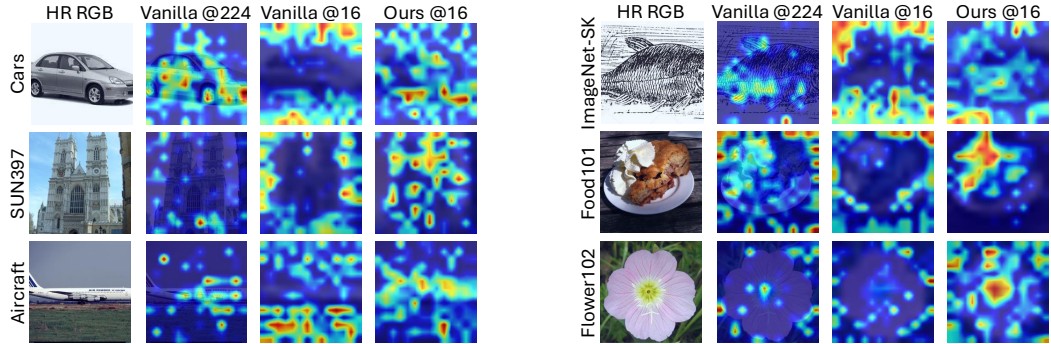

Figure 23: **Effect of LR token**: '@' is input resolution. Vanilla model attention is scattered at $16\times16$ (compared to $224\times224$), while our LR tokens focus on the object, capturing fine-grained details.

Furthermore, we present a side-by-side comparison between the LR token introduction for WAR-16 and SAR-16 metrics in **Figure 22**.

**Spearman correlation for other resolutions:** In **Figure 24**, weights derived for $16\times16$ are used for other models. Weights for $16\times16$ hold for $32\times32$ but degrade for $64\times64$ and $128\times128$ becoming identical to SAR. **Figure 25** shows different configurations for obtaining dataset weights.

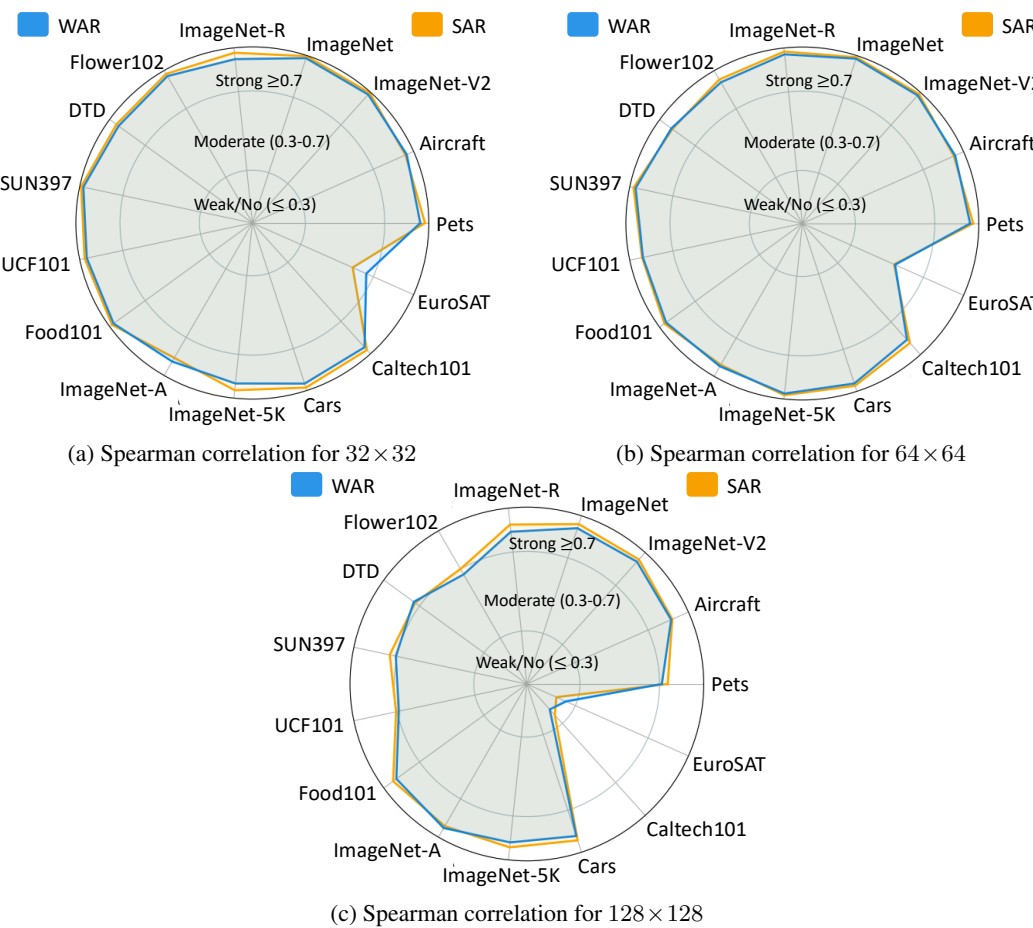

(a) Spearman correlation for $32 \times 32$

(b) Spearman correlation for $64 \times 64$

(c) Spearman correlation for $128 \times 128$

Figure 24: **Spearman Correlation** for weights derived for $16 \times 16$ for higher resolutions.

**Samples of Diffusion Generated Images:** In **Figure 26** and **Figure 27**, we showcase a few sample images generated using PIXART-$\alpha$. These plots are an extension of Figure 11 presented in the main paper.

## H   More Observations

**Semantically correct mispredictions:** As described in Figure 2 of the main paper, misclassified low-resolution images are still assigned reasonable semantic predictions. Here in **Figure 28**, we showcase more such examples where the above phenomenon holds.

**Real World low-resolution images:** We have taken a few real-world low-resolution sample images from Google as shown in **Figure 29** to see the considered model's performance. Here, we have considered the top-5 predictions of the model and see which indicates (i) correct predictions, (ii) semantically reasonable predictions, and (iii) wrong predictions. The ground labels (or templates) for considered images are chosen from Imagenet.

## I   Limitation

A key limitation of our study is the lack of detailed analysis of the pre-training datasets, which could provide deeper insights into model performance, particularly regarding how dataset quality impacts robustness. However, due to the scale and unavailability of certain datasets, conducting such an analysis is challenging, though it remains a promising direction for future work with available resources and accessible datasets.

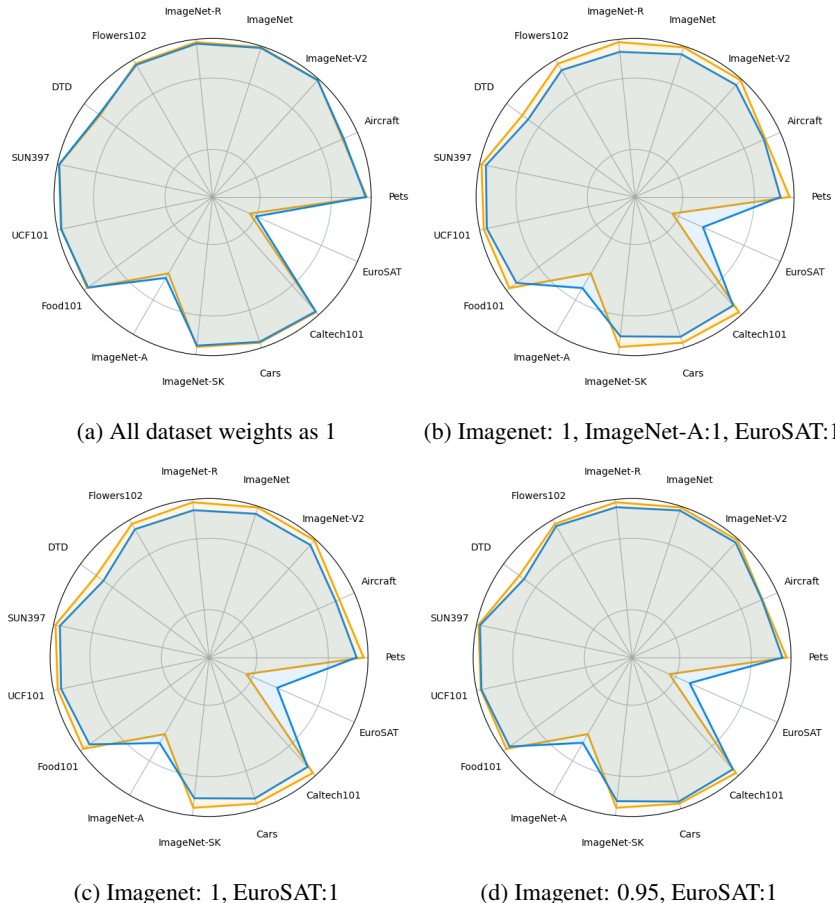

(a) All dataset weights as 1

(b) Imagenet: 1, ImageNet-A:1, EuroSAT:1

(c) Imagenet: 1, EuroSAT:1

(d) Imagenet: 0.95, EuroSAT:1

Figure 25: **Spearman Correlation** for different optimization function. The optimization objective is to maximize the mentioned dataset Spearman correlation (SC). For Example 'Imagenet: 0.95, EuroSAT:1' means: SC (Imagenet)$\times 0.95$ + SC (EuroSAT)$\times 1$.

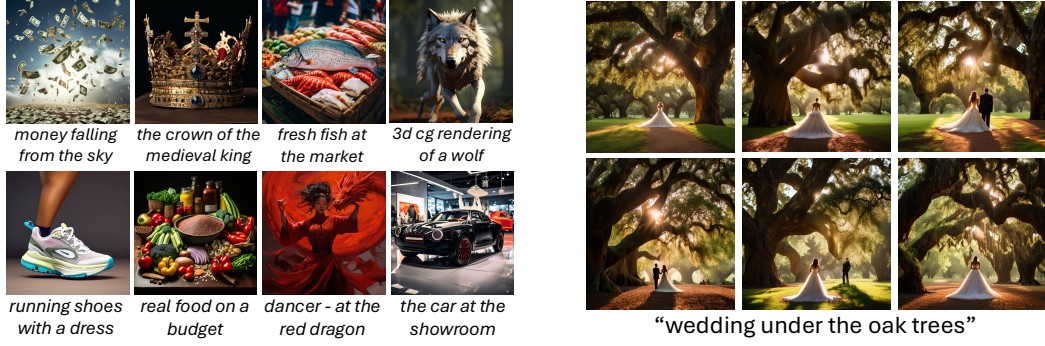

Figure 26: **Synthetic Images**: Images generated using PIXART-$\alpha$ (Chen et al., 2023a) using the captions randomly sampled from Conceptual Captions (Sharma et al., 2018). *Left*: Sample Images, while *right* shows multiple images per caption generated via different seeds. More examples of Figure 11 (in main paper).

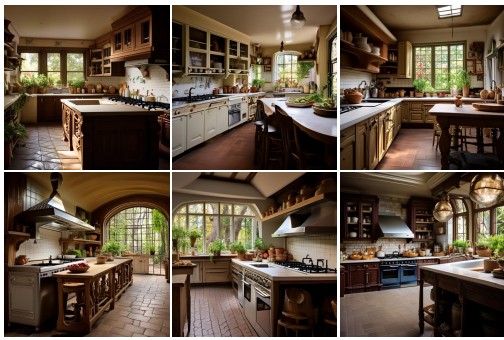 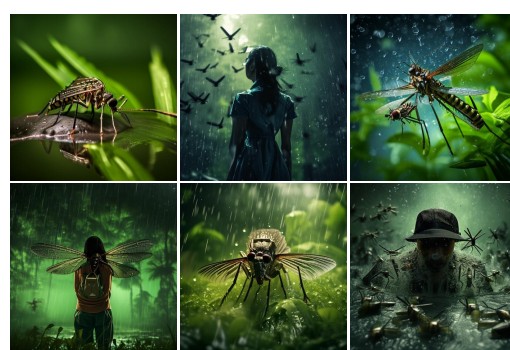

"photo of second kitchen in the house"

safety from mosquitoes is important in this monsoon .

Figure 27: **Synthetic Images2**: Mutliple Images / Caption. More examples of Figure 11 (in the main paper)

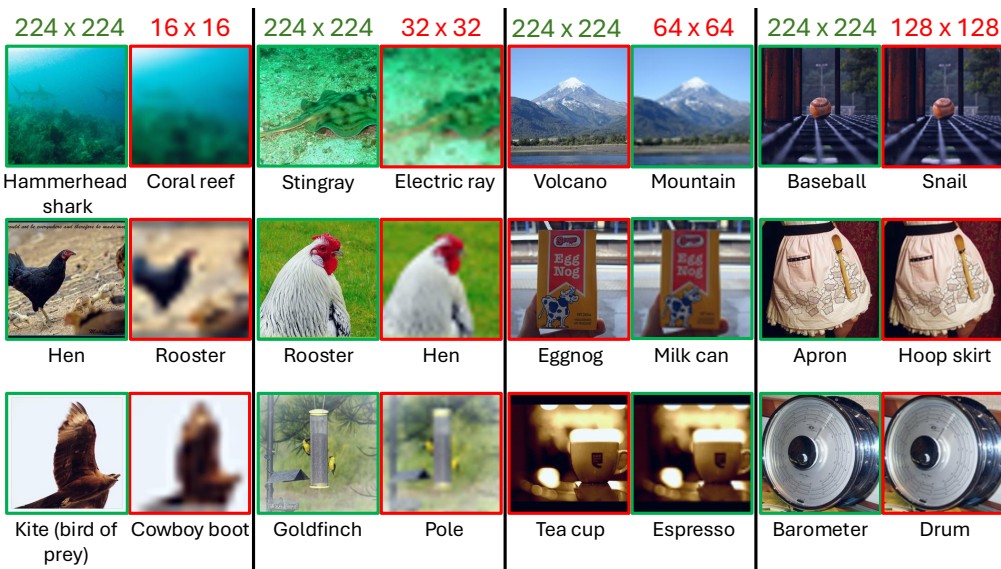

Figure 28: **Semantically Correct Predictions**: More examples of Figure 2 in the main paper. Visually different examples were chosen to show the usefulness of the pre-trained weights even in low resolution.

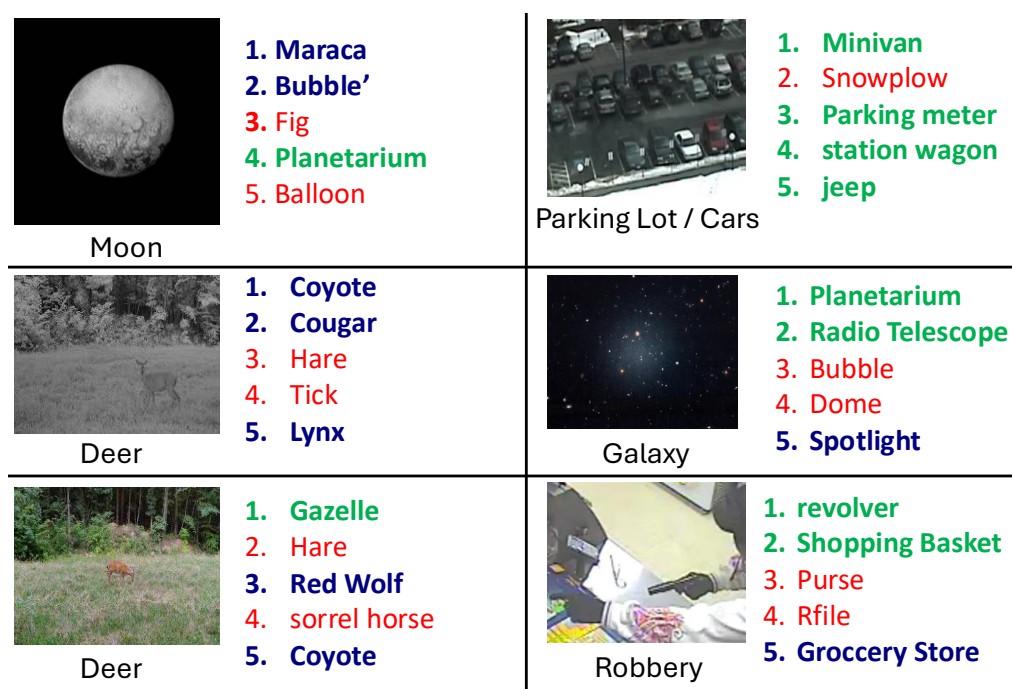

Figure 29: **Real World low-resolution images**: Top-5 predictions for images taken from Google (true label shown, below image). **Blue** indicates Semantically reasonable prediction, **Green** indicates correct prediction, and **Red** means wrong prediction. EVA-02-CLIP-B/16 model predictions for unknown resolution (real-world footage). Labels/templates are chosen from the **ImageNet dataset**.