# OpenReview forum: "LR0.FM: Low-Res Benchmark and  Improving robustness for Zero-Shot Classification in Foundation Models"
_ICLR.cc/2025/Conference — ICLR 2025 Poster_

### Official Review · Reviewer_TJWB · 2024-11-01

**Soundness:** 3
**Presentation:** 3
**Contribution:** 3
**Rating:** 8
**Confidence:** 4

**Summary:**

The paper proposes a low-resolution benchmark named LR0.FM and a simple training method, LR-TK0, can improve low-resolution performance. The paper measures the low-resolution performance of vision-language foundation models using LR0.FM and presents valuable findings based on performance analysis. LR-TK0 suggests adding a visual prompt to each spatial token of every layer to enhance the low-resolution performance of VL models. Experiments exhibit the effect of LR-TK0.

**Strengths:**

- LR0.FM benchmark includes diverse VL foundation models on various datasets. I believe the low-resolution reports on 10 foundation models across 66 backbones for 15 datasets might be helpful to a lot of researchers in VL foundation models.
- The paper includes meaningful revision on evaluation metrics, i.e., `improved relative robustness` and `weighted aggregation for resolution.` Even though these metrics are not perfect, I think raising issues on metrics and trying to solve them can provide positive guidance for the benchmark field.
- The paper provides various aspects of benchmark results in Figures 5 - 7, which are more impressive and insightful than simple ranking.

**Weaknesses:**

- Although I agree with the efficiency of the low-resolution benchmark, I'm not sure of the value of low-resolution performance in the vision-language model domain. Generally, images in `16x16` or `32x32` are not the main target of VL models, which limits the contribution and impacts of LR0.FM benchmark.

- There is no comparison with other benchmark results. It would be more informative if the paper provided whether the low-resolution robustness aligns with other robustness benchmarks or has a unique pattern.

- LR-TK0 is a valid way to improve low-resolution performance. But, the connection between LR0.FM and LR-TK0 is not strong. It feels like reading two related papers rather than enhancing the contribution of LR0.FM.

**Questions:**

- Is the low-resolution performance important for vision-language models? Do you have any practical use case on it?

- Compared to another robustness benchmark in VL models, does low-resolution robustness show a different pattern, or is it similar to others?

- Problem A and Problem B seem to be practical issues for benchmarking. But I'm not sure how significant these problems are. Could you provide a quantitative report on the significance of this problem?

- In Section 4, the description points to Figure 5 (right, (a)) and (right (b)). But, only (i) and (ii) exist in Figure 5 (right).

---

> ### Author Response · Authors · 2024-11-20
> **Rebuttal Response 1**
>
> **W1: Although I agree with.... LR0.FM benchmark.**
> &
> **Q1: Is the low-resolution performance .... practical use case on it?**
>
>
> One of the reasons zero-shot / open-vocabulary models are gaining popularity is because of their real-world application (e.g. YOLO-WORLD [1]). But the real world is noisy and has low-resolution images, which is where the need for robustness arises.
> 16x16 and 32x32 do not only represent the image sizes (small face, far away galaxies) `[Line 52 & 69]` but also represent pixelation, i.e. enlarging 16 x 16 to 224 x 224 (or higher resolution) mimics far away objects like HQ surveillance camera footage for far object `[Line 192-194]`, or privacy protected data `(Line 52)`. `Figure 29`, contains some real-world examples where the model makes simple mistakes in recognizing images if the images contain low-resolution artifacts like pixelation.
>
> [1]: Cheng, Tianheng, et al. "Yolo-world: Real-time open-vocabulary object detection." Proceedings of the IEEE/CVF Conference on Computer Vision and Pattern Recognition. 2024.
>
> ----
> ----
>
> **W2: There is no comparison.... or has a unique pattern.**
>
> To the best of our knowledge, there isn't any related work for low resolution in zero-shot conditions for foundation models (FMs/VLMs) `[Line 150-151]`.
> The closest work to ours is Robust SAM [1], which tries to improve the robustness of the Segment Anything Model (foundation model) against noisy images. We have tried modifying the segmentation model to our classification task (denoising modules, and trainable prompts) in `Table 4`. Real-world benchmarking results that overlaps with some of our findings include:
> 1) Noisy Image Segmentation masks [1]: Preserving original weights helps develop robustness `(LR-TK0)`, and Fine-tuning drops the zero-shot capabilities `(Figure 7 left)`.
> 2) ImageNet-1K pretraining may be more robust than LAION-2B pre-training (in case of fine-tuning). [2] `(Figure 6 left)`.
>
> [1] Wei-Ting Chen, Yu-Jiet Vong, Sy-Yen Kuo, Sizhou Ma, and Jian Wang. Robustsam: Segment anything robustly on degraded images. 2024.
> [2] Hwang, Jaedong, et al. "ImageNet-RIB Benchmark: Large Pre-Training Datasets Don't Guarantee Robustness after Fine-Tuning." arXiv preprint arXiv:2410.21582 (2024).
>
> -----
> -----
>
> **W3: LR-TK0 is a valid way .... enhancing the contribution of LR0.FM.**
>
> Our proposed LR-TK0 is dependent on LR0.FM in some fundamental ways:
> 1) Robustness (evaluation metric) improvement is derived using the benchmarking of 66 models (WAR metrics)
> 2) Knowledge of `(Figure 7 (right)` indicates that the lower / shallow layer deviates from HR model layers more than the deeper layer.  This contradicts traditional LR-based techniques that deploy trainable modules on top of the final features, focusing more on the deeper layers than the initial ones. `[Line 373-374]`
> 3) Knowledge of `Figure 2`, which indicates models make semantically correct predictions on low resolution, is the motivation for preserving pre-trained weights.
>
> Additionally, compared to the research in low resolution (far more comprehensive), our solution is very basic and intended to motivate future research in this direction.
> Existing techniques like denoising module [1], inverse LR training module [2], and specialized resolution modules [3], if deployed on  LR  Tokens will likely see a far superior performance gain. Our solution is a basic setup for their solutions.
>
> [1] Wei-Ting Chen, Yu-Jiet Vong, Sy-Yen Kuo, Sizhou Ma, and Jian Wang. Robustsam: Segment anything robustly on degraded images. 2024.
> [2]: Lu, Yuhang, and Touradj Ebrahimi. "Cross-resolution Face Recognition via Identity-Preserving Network and Knowledge Distillation." 2023 IEEE International Conference on Visual Communications and Image Processing (VCIP). IEEE, 2023.
> [3] Chai, Jacky Chen Long, et al. "Recognizability embedding enhancement for very low-resolution face recognition and quality estimation." Proceedings of the IEEE/CVF Conference on Computer Vision and Pattern Recognition. 2023.
>
> -----
> -----

---

> > ### Author Response · Authors · 2024-11-20
> > **Rebuttal Response 2**
> >
> > **Q2: Compared to another.... similar to others?**
> >
> > We have intentionally steered clear from overlapping our analysis with existing works. The basic overlap with the existing world includes:
> > 1) [1] Accuracy drops with severity `(Figure 1)`, models are robust to most types of distribution shifts except compression `(low resolution)`, CNN < Transformer `(Supplementary Page 21, Section D)`, Collapse of t-SNE features on compression `(Figure 8)`.
> > 2) [2] Preserving original weights helps to develop robustness `(LR-TK0)`, Fine-tuning drops the zero-shot capabilities `(Table 5)`
> > 3) [3] Robustness increases with model size `(Figure 5 (right) i)`
> > 4) [4] ImageNet-1K pretraining may be more robust than LAION-2B pre-training (in case of fine-tuning). `[Line 306-310]`
> >
> > [1]  Schiappa, Madeline Chantry, et al. "Robustness Analysis on Foundational Segmentation Models." Proceedings of the IEEE/CVF Conference on Computer Vision and Pattern Recognition. 2024.
> > [2] Wei-Ting Chen, Yu-Jiet Vong, Sy-Yen Kuo, Sizhou Ma, and Jian Wang. Robustsam: Segment anything robustly on degraded images. 2024.
> > [3] Michaelis, Claudio, et al. "Benchmarking robustness in object detection: Autonomous driving when winter is coming." arXiv preprint arXiv:1907.07484 (2019).
> > [4] Hwang, Jaedong, et al. "ImageNet-RIB Benchmark: Large Pre-Training Datasets Don't Guarantee Robustness after Fine-Tuning." arXiv preprint arXiv:2410.21582 (2024).
> >
> > ----
> > ----
> >
> > **Q3: Problem A and Problem B seem ....  on the significance of this problem?**
> >
> > We appreciate the reviewer for understanding the importance of both the problems of robustness evaluation metrics.
> > `Supplementary Table 9` has quantitative values for abnormally high robustness scores `(Problem A)`. Culprits models include ALBEF and BLIP which constitutes 12 (4 + 8) / 66 backbones used in the analysis, across EuroSAT, FVG Aircraft, and Stanford cars (3 / 15) datasets. `Problem A` is not an outlier, but a significant issue.
> >
> > `Problem B` becomes more relevant when comparing models across datasets. For our analysis and ablation, we have mostly used WAR metrics. All it tells that patterns and relative comparison include all 15 datasets, which otherwise actually be only 14 datasets if SAR scores were reported as it doesn't correlate with EuroSAT.
> >
> > `Figure 4` shows:
> > i) WAR increases the correlation score of EuroSAT (0.26 → 0.49) from weakly / no correlation to moderate correlation, and ImageNet-A (0.56 → 0.68, strong correlation is 0.70).
> > ii) For highly correlated datasets small difference between SAR and WAR is ideal i.e. WAR preserves the high correlation scores of these datasets.
> >
> > ----
> > ----
> >
> > **Q4: In Section 4, the description... exist in Figure 5 (right).**
> >
> > Thank you for bringing these to our attention. These changes have been added to the revised edition.
> >
> >
> > ----
> > ----
> >
> > We will be happy to answer any more doubts and would thank you once again for the feedback.

---

> ### Comment · Reviewer_TJWB · 2024-11-21
>
> Thank you for your response.
>
> It addressed my questions.
>
> I believe weaknesses 1 and 3 cannot be fully resolved through rebuttal comments.
> However, despite these weaknesses, the paper offers valuable contributions.
>
> I will maintain my initial rating.
>
> For W2 and Q2, I hoped to see benchmark comparisons similar to [1].
> Including such comparisons across various benchmarks, as in [1], could enhance the paper's contribution.
>
> [1] Do Better ImageNet Models Transfer Better?, CVPR 2019

---

> > ### Author Response · Authors · 2024-11-24
> > **Reply: Reviewer TJWB**
> >
> > We appreciate the reviewer's prompt response. We are glad we were able to address your questions.
> >
> > Regarding W2 and Q2; thank you for referring us to [1]. We agree with the reviewer that comparisons across various benchmarks, as in [1], could enhance the contribution. Most of our analysis across various benchmarks is provided in the supplementary `(Figure 17, Figure 18, and Figure 19)` and main submission `(Figure 5, figure 6, Figure 8)`. It will be challenging to provide all the analysis similar to [1] as we focus on zero-shot foundation models which don't share training protocols (architectures, and pretraining) for us to be able to single out such factors that help robustness.
> >
> > We can include the histogram similar to Figure 6 in [1] to compare models across datasets and also convergence analysis for LR-TK0 (similar to Figure 8 in [1]).
> >
> >
> >
> >
> > If the reviewer recommends any other analysis, we will try our best to include it.
> >
> > [1] Do Better ImageNet Models Transfer Better?, CVPR 2019

---

### Official Review · Reviewer_d8wv · 2024-11-01

**Soundness:** 2
**Presentation:** 2
**Contribution:** 2
**Rating:** 6
**Confidence:** 4

**Summary:**

This paper proposes a benchmark LR0.FM for foundation models on LR. The authors propose the metric WAR-n to measure the robustness of foundation models. Additionally, the authors propose a method LR-TK0 to enhance performance on LR.

**Strengths:**

1. I believe this benchmark for LR is important.

2. Extensive benchmark experiments analyses about LR0.FM.

**Weaknesses:**

1. In Problem B) SAR overlooks datasets, it is unclear what Spearman Rank correlation and AX optimization are, as well as how to obtain weights through AX optimization, making it difficult to understand why WAR is proposed.

2. In Table 2, Ours has performance improvement over EVA-B/16, which should be attributed to fine-tuning using the synthetic images&LR images and it certainly improves the performance. Therefore, the experiments comparison are not fair and please explain what the appealing novelty is. Additionally, in Table 3, ours performs better than the SR method, but it is unclear what the differences are between ours and SR methods and why ours is more effective.

3. In the introduction, for "We observe that existing robustness metrics have some limitations", you should introduce what the existing metrics are.

4. In Figure 1, it's not clear what the y-axis is.

5. In section 3, in "The naming convention would represent the publicly available backbone pretrained weight names. Pre-training Sizes are indicated like CLIP-ViT H(400M), where M denotes million image-text pretraining.", "CLIP-ViT H" is not the weight names and it should be clear that M is the size of dataset.

6. In evaluation metrics, in $\gamma_n^D$, only 224 or each HR is used? May be $\gamma_n^D = 1 - \left(\frac{A_{m} - A_n^D}{A_{m}}\right), m=224,256,378...$ is better.

7. There are some wrong details. Here, only the obvious mistakes are pointed out: In figure 10, "Fire (& ice) icons represent trainable (& frozen) parameters." is not a suitable caption and there are no (a) and (b) in the figure

8. In table 5, what is the difference between Baseline row and the next below row?

9. In Grad-CAM results, it's not clear what the vanilla model is.

10. The figure caption should explain the figure content rather than the meaning of the axes. Several figures have this issue, such as Figure 13 and Figure 14.

11. In some figures, it's not needed to use arrow, like in figure 4 and figure 5.

**Questions:**

Please respond to the questions in the weakness.


____________________
post-rebuttal message:

The authors have promised the suggested revisions, which may facilitate the readers' understanding.

Therefore I have raised my rating from 3 to 6, accordingly.

---

> ### Author Response · Authors · 2024-11-20
> **Rebuttal Response 1**
>
> **W1: In Problem B) SAR ... understand why WAR is proposed.**
>
> Apologies for the confusion regarding the proposed metrics. We will try to simplify the ***“Problem B of SAR overlooks dataset”*** and have updated the simplification to the revised copy as well `[Line 248-269]`.  We will try to answer the concerns in parts
>
> Once robustness is calculated (using traditional $\gamma_{n}^{D}$ or our improved one $\Gamma_{n}^{D}$), a common solution is to average scores across the datasets (giving each dataset a score of 1) this is called ***SAR, Simple Aggregated Robustness*** `[Line 198-200]`. SAR is a simple averaging of robustness scores across all the datasets.
>
> **1) Ordering of the model**
> Once the robustness scores are aggregated across the dataset, the standalone aggregated scores don't mean anything unless we are comparing models. The ranking of the models after aggregation across the datasets compares the models. `[Line 247-249]`.
> Ideally, model ranking, after averaging robustness across datasets, should stay consistent with their rankings on individual datasets. This consistency of ranking is measured by ***Spearman Rank correlation*** (measures relative ordering of two distributions).
>
> **Middle subfigure of Fig 4**
> Whether we use traditional relative robustness $\gamma_{n}^{D}$ (abnormally high value) or our improved robustness $\Gamma_{n}^{D}$ `(Problem A solution)`, the SAR ranking of models is uncorrelated / weakly correlated with EuroSAT ranking of models (0.26) and moderately correlated with ImageNet-A (0.56) `[Line 249-251]`. WAR adjusts the dataset weights so that the model rankings after aggregation reflect each dataset fairly `(fig. 4 (right))`.
>
> **2) Objective function of AX optimization**
> AX optimization is a hyperparameter optimization tool that maximizes a certain objective. In our case, adjust the weights of the datasets (hyperparameters) with the overall goal of maximizing the Spearman Rank correlation between the final model ranking obtained after the weighted averaging of datasets and individual dataset ranking on empirically found objective function `(equation 2)`
> $$0.95 \times SC(Imagenet) + 0.95 \times  SC(ImageNet-V2) + 0.95 \times  SC(DTD) + SC(ImageNet-A) + SC(EuroSAT)$$
>
> `Figure 4` indicates this via a spider curve.
> 1) WAR increases the correlation score of EuroSAT (0.26 → 0.49) from weakly / no correlation to moderate correlation, and ImageNet-A (0.56 → 0.68, strong correlation is 0.70).
> 2) For highly correlated datasets small difference between SAR and WAR is ideal i.e. WAR preserves the high correlation scores of these datasets.
>
> -----
> -----
>
> **W2: In Table 2, Ours .... why ours is more effective.**
>
> We apologize for the confusion and `Table 2` caption is updated to reflect ***LR-TK0 improvement on various Foundation models***. `Table 2` is not a comparison with the baseline, hence “fairness” is not applicable. We have tried to show the gain in robustness by applying our LR-TK0 technique to the baseline foundation models.
>
> There are two novelties here [Line 360-364]:
> 1) ***We are trying to increase the robustness of the models against LR images, without discarding the pre-trained weights with minimal parameter gain***. `Table 2`: 224 x 224 accuracy indicates we are pretty much retaining the initial zero shot accuracy (1-2% accuracy drop). For LR 16 x 16, the accuracy jump is increasing from (4-9%) across different models.
> 2) ***We are not training/fine-tuning on any of our target datasets, making it data-free (synthetic data)*** (Unlike previous methods, RobustSAM [1]). Our goal is to propose a “zero-shot” technique for increasing robustness against LR.
>
> `Table 3` SR methods & `Figure 9`: Both intend to show SR (super-resolution) methods don't work well in zero-shot conditions, i.e. they need target datasets to improve performance, which defeats the zero-shot aspect of LR robustness. Our method trained on synthetic datasets doesn't require finetuning on any of the target datasets.
>
> [1] Wei-Ting Chen, Yu-Jiet Vong, Sy-Yen Kuo, Sizhou Ma, and Jian Wang. Robustsam: Segment anything robustly on degraded images. 2024.
>
> ----
> ----
>
> **W3: In the introduction, ... the existing metrics are***.
>
> Thank you for the suggestion. The revised version now reads; “Metrics for measuring robustness ($\gamma$, Schiappa et al. (2024)) and its averaging across datasets (SAR) have some limitations;“ `[Line 78-80]`
>
> ----
> ----
>
> **W4: In Figure 1, it's not clear what the y-axis is.**
> Apologies for the confusion. The y-axis is zero shot classification accuracy and the x-axis is the resolution. `Figure 1` caption now reflects this.

---

> ### Author Response · Authors · 2024-11-20
> **Rebuttal Response 2**
>
> **W5: In section 3, in ....... M is the size of dataset.**
>
> Thank you for bringing these to our attention. The revised version now reads; `[Line 175-177]`
> Backbones are referred to using their publicly available pre-trained weights, e.g. CLIP-ViT L (400M), which means: CLIP model ViT-L architecture, pre-trained on 400 million datasets.
>
> ----
> ----
> **W6: In evaluation metrics, in $\gamma_n^D$.... m=224,256,378… is better.**
>
> Thank you for the constructive suggestion, it's now included in the revised version at `[Line 195-197]`.
>
> ----
> ----
> **W7: There are some.... no (a) and (b) in the figure**
>
> Thank you for bringing these to our attention. All corrections are added to the revised copy. The final copy has replaced (a) with “left” and (b) with “right” in `Fgure 10 caption`.
>
> ----
> ----
>
> **W8: In table 5, what is .... next below row?**
>
> Apologies for the confusion. The baseline row now reads as ‘Baseline (frozen)’ which is pretrained EVA-B/16 zero-shot performance. The next row is *“not”* frozen, (end-to-end finetuning), *“doesn't”* have LR-tokens (our contribution), and *“doesn't”* have a task-specific classifier (similar to our technique). `[Line 481-483]`.
>
> ----
> ----
>
> **W9: In Grad-CAM results, it's not clear what the vanilla model is.**
>
> Thank you for bringing these to our attention. `Figure 16 (& caption)` now reads baseline instead of vanilla, where EVA-B/16 is the baseline model `[Line 431]`.
>
> ----
> ----
>
> **W10: The figure caption should....as Figure 13 and Figure 14.**
>
> Thank you for bringing these to our attention. All captions are updated to include more descriptions.
>
> ----
> ----
>
> **W11: In some figures, it's not needed to use arrow, like in figure 4 and figure 5.**
>
> Thank you for bringing these to our attention. The arrows have been removed in the final copy.
>
> ----
> ----
>
> We hope that we have resolved all the issues raised by the reviewer. We are happy to answer any more doubts and would thank you once again for the feedback.

---

> ### Author Response · Authors · 2024-11-25
> **Feeback on rebuttal**
>
> Dear reviewer,
>
> Thank you once again for your time in reviewing our paper and providing valuable feedback. As the discussion period extended till 2nd December, we are reaching out to see if you have any further questions or pending issues.
>
> We have hoped to resolve your concerns with WAR-SAR evaluation metrics, and clarity in writing.
>
> Please let us know if you have any follow-up comments or require additional clarifications.

---

> ### Comment · Reviewer_d8wv · 2024-11-29
> **Rating is updated.**
>
> Thanks for the detailed response.
>
> Please update the final version accordingly.
>
> The rating has been updated from 3 to 6.

---

> > ### Author Response · Authors · 2024-11-29
> > **Reply: Reviewer d8wv**
> >
> > We thank the reviewer for their response and are glad we were able to address their concerns. We will be happy to answer any remaining questions.

---

### Official Review · Reviewer_pEiF · 2024-11-02

**Soundness:** 3
**Presentation:** 2
**Contribution:** 3
**Rating:** 6
**Confidence:** 4

**Summary:**

This paper has explored the zero-shot ability of the visual-langrage foundation model on low-resolution images. The first contribution is that a new robust metric is proposed, named weighted aggregated robustness, which takes into account the cues of the order of the datasets and the gap between the best and random performance on the individual dataset. With abundant experiments considering different kinds of foundation models, backbones, and datasets, the paper reveals that the model is more robust with a larger size and better training data quality. Based on these analyses, a new strategy is designed to improve the zero-shot performance of the foundation model on low-resolution images by adding low-resolution tokens.
This research work is interesting and the paper gives abundant experiments and analyses. However, the illustration and description of the figures, the structure of the paper, and the clarity of some core concepts make the paper hard to follow.

**Strengths:**

1. The new robustness metric is reasonable.
2. The experiment analysis is sufficient.
3. The effectiveness of the proposed LR-TK0 is verified.
4. The viewpoint of this paper is interesting.

**Weaknesses:**

1. The "misleading high robustness" of SAR is clear. The "SAR overlooks dataset" is hard to understand as: 1) What does the ordering of the model mean? 2) What is the object function to optimize the model weight? 3) From the middle subfigure of Fig 4, the difference between SAR and WAR is small. how does the figure prove the SAR overlooks datasets? 4) From the last sentence of page 5, both SAR and WAR are used. SAR is for the individual dataset and WAR is used across the datasets. Does it mean that it does matter that SAR has  "misleading high robustness" problem? 5) How to choose the hyperparameter alpha in Eq. 1?
2. The figures are hard to read as:1) Too many factors are shown with the area of the circle (Fig. 5), the length of the bar (Fig. 7), the color of the text (Fig. 3), etc. 2) The subfigures has no relation in some figure, such as Fig. 3 to Fig. 5.
3. The paper claims that 66 backbones are used. What are they?
4. The LR-TK0 technique is not clear on page 7 as 1) it lacks the introduction of LR token, 2) What's "Path generation" in Fig. 10? 3)How to compute the contrastive loss as there are three numbers in Fig. 10?
5. The experiment analyses of Table 2 and Table 3 are not clear. What's the experimental setting when the SR method is used in Table 2? It lacks a summary of Table 3.
6. Some minor typos: 1)The caption of Table 2. 2)The format of the figure references is not consistent. 3) "The naming convention ..." in Line 178 is hard to understand. 4) There's an extra period after "Problem B) SAR overlooks datasets".

**Questions:**

My main concern is the weakness 1 and 4 in the weaknesses part. I think it could not be clearly solved due to the page limitation. I suggest the authors choose the core and important parts, then make them clearly introduced and put them in the main body.

---

> ### Author Response · Authors · 2024-11-20
> **Rebuttal Response 1**
>
> **W1: The "misleading high robustness" ....  5) How to choose the hyperparameter alpha in Eq. 1?**
>
> Apologies for the confusion regarding the proposed metrics. We will try to simplify the “Problem B of SAR overlooks dataset” and have updated the simplification to the revised copy as well `[Line 248-269]`.
>
> **4) Confusion regarding SAR**
> We have 15 datasets, one model may be robust on 1 dataset and worse on the other. This robustness is measured by traditional relative robustness $\gamma_n^D$ and can give abnormally high values for certain models on certain datasets. `(Problem A, [Line 202-206])`. Problem A solution is ***improved relative robustness*** represented by $\Gamma^{D}_{n}$ that won't give an abnormally high value.
>
> Once robustness is calculated (traditional $\gamma_n^D$ or our improved one $\Gamma^{D}_{n}$), how do we compare the robustness of the two models? Dataset by dataset basis is not feasible. A common solution is to average scores across the datasets (giving each dataset a score of 1) this is called ***SAR, Simple Aggregated Robustness*** `[Line 198-200]`. SAR is not dataset-specific, but a simple averaging of robustness scores across all the datasets.
>
> **1) What does the ordering of the model mean**
> Once the robustness scores are aggregated across the dataset, the standalone aggregated scores don't mean anything unless we are comparing models. The ranking of the models after aggregation across the datasets compares the models. `[Line 247-249]`.
> Ideally, model ranking, after averaging robustness across datasets, should stay consistent with their rankings on individual datasets. This consistency of ranking is measured by ***Spearman Rank correlation*** (measures relative ordering of two distributions)
>
> **3) middle subfigure of Fig 4**
> Whether we use traditional relative robustness $\gamma_n^D$ (abnormally high value) or our improved robustness $\Gamma_n^{D} $ `(Problem A solution)`, the SAR ranking of models is uncorrelated/ weakly correlated with EuroSAT ranking of models (0.26) and moderately correlated with ImageNet-A (0.56) `[Line 249-251]`. `Figure 4` middle doesn't show WAR metrics. $\Gamma^{D}_{16}$ is the “improved robustness metrics”, `(solution to Problem A)`. WAR adjusts the dataset weights so that the model rankings after aggregation reflect each dataset fairly `(fig. 4 (right))`.
>
> **2) Objective function of AX optimization**
> AX optimization is a hyperparameter optimization tool that maximizes a certain objective. In our case, adjust the weights of the datasets (hyperparameters) with the overall goal of maximizing the Spearman Rank correlation between the final model ranking obtained after the weighted averaging and individual dataset ranking on empirically found objective function `(equation 2)`.
> $$0.95 \times  SC(Imagenet) + 0.95 \times SC(ImageNet-V2) + 0.95 \times SC(DTD) + SC(ImageNet-A) + SC(EuroSAT)$$
>
> **5) How to choose alpha**
> Alpha is the rate at which robustness declines as accuracy approaches random prediction. Any alpha >> 1 is a good choice, `(Figure 21)`. We chose 200  as a middle between 100 (the drop starts at 0.2) and 500 (the drop starts very close to 0). `[Line 246-247]`
>
> -----
> -----
>
> **W2: The figures are hard to.....such as Fig. 3 to Fig. 5.***
>
> We apologize for the lack of clarity in the figure. As a revision, we have increased the spacing between figures for easy readability.
> 1) `Figure 5 (right) i)` size shows GFLOPs non-correlation with robustness, but model size correlates with robustness. `Figure 5 (right) ii)` size indicates model size (positively impact robustness).
> 2) `Figure 7 (mid)` The length of the bar is indicative of the average robustness of (66 models) for the dataset.
> 3) `Figure 3` word cloud is generally used in benchmarking papers for summarizing datasets, (also used in Elevater [1])
> 4) Subfigures (different goals) save us white space that would be created with individual figures.
>
> [1] : Elevater: A benchmark and toolkit for evaluating language-augmented visual model (Nuerips’22)
>
> -----
> -----
>
> **W3: The paper claims that 66 ... What are they?**
>
> The revised version contains the explicit mention of the 66 backbones in `Table 1` caption.
> `Table 1` provides a comprehensive overview of all the foundation models considered for this study and their corresponding backbones (66 in total). For example `(in row-1 of Table-1)`, the CLIP model has four Vision Transformers (ViTs) and five ResNet backbones.

---

> > ### Comment · Reviewer_pEiF · 2024-11-28
> >
> > Thanks for the author's response and modification of the manuscript. Most of my concerns are solved. I will raise the score.

---

> > > ### Author Response · Authors · 2024-11-28
> > > **Reply: Reviewer pEiF**
> > >
> > > We thank the reviewer for their response and are glad we were able to address most of their concerns. We will be happy to answer any remaining questions.

---

> ### Author Response · Authors · 2024-11-20
> **Rebuttal Response 2**
>
> **W4: The LR-TK0 technique ...  three numbers in Fig. 10?**
>
> We apologize for the lack of clarity on certain terminology.
> 1) LR tokens are just ordinary trainable transformer tokens, indifferentiable from any other transformer tokens added on top of spatial tokens `[Line 367-370]`. Papers often name these “newly added tokens”, for example, “Class token in ViT” [1], “Distillation token” in Deit [2], “Prompts” in VPT [3].
> 2) There is no “Path generation", but “Patch Generation” in `Figure 10`. Patch Generation is the stage before transformer blocks where RGB image patches are converted into patch tokens.
> 3) Apologies for the unclarity in the original figure. `(Figure 10 right)` now has been fixed to show two contrastive losses which takes two embeddings to align them contrastively.
>
> [1]: Alexey Dosovitskiy. An image is worth 16x16 words: Transformers for image recognition at scale. International Conference on Learning Representations, 2021.
> [2]: Touvron, Hugo, et al. "Training data-efficient image transformers & distillation through attention." International conference on machine learning. PMLR, 2021.
> [3]: Menglin Jia, Luming Tang, Bor-Chun Chen, Claire Cardie, Serge Belongie, Bharath Hariharan, and Ser-Nam Lim. Visual prompt tuning. In European Conference on Computer Vision, pp. 709–727. Springer, 2022.
>
> -----
> -----
>
> **W5: The experiment analyses of .... lacks a summary of Table 3.**
>
> We apologize for the non-clear description.
>
> **Table 2**
> `Table 2` caption explicitly mentions LR-TK0 for different foundation models. It doesn't have SR (super-resolution) methods, nor our method LR-TK0 (as shown in `Figure 10`) uses any super-resolution. `Table 2` has baseline foundation models and improvement via our LR-TK0 technique.
> `[Line 467-471]` describes the analysis of `Table 2`, indicating consistent enhancement of robustness at low resolutions (16×16 & 32×32), particularly for MetaCLIP. While the low resolution is often seen as a domain shift problem [1], leading to potential declines in HR performance, our multi-scale training and HR teacher distillation minimize accuracy drops at higher resolutions.
>
> **Table 3**
> `[Lines: 472-476]` describes “`Table 3` compares EVA-B/16 with super-resolution (SR) methods, with SR methods performing poorly in zero-shot settings for very low resolutions `(fig. 9)`. In contrast, our approach is better suited for zero-shot scenarios. Diffusion-based SR method like IDM is too computationally expensive to evaluate on large datasets like ImageNet (results in Supplementary `(Section F2 and F3)`)”.
> Regarding the experimental settings of the SR method in `Table 3`, we provide the details along with a code snapshot in the `supplementary Section-E [lines: 1137-1174]`.
>
> [1] Efficient Low-Resolution Face Recognition via Bridge Distillation. IEEE Transactions on Image Processing, 2020.
>
> -----
> -----
>
> **W6: Some minor typos: ...  SAR overlooks datasets".**
>
> Thank you for bringing these to our attention. All corrections are made to the revised version.
> We have followed the convection: If a sentence starts with word "Figure" (capital F) and if it mentions it in the middle of the sentence "fig." (small f).
> We have reworded the naming convention for ease of understanding `[Line 175-177]`:
> Backbones are referred to using their publicly available pre-trained weights, e.g. CLIP-ViT L (400M), which means: CLIP model ViT-L architecture, pre-trained on 400 million datasets.
>
> -----
> -----
>
> We will be happy to answer any more doubts and would thank you once again for the feedback.

---

> ### Author Response · Authors · 2024-11-25
> **Feeback on rebuttal**
>
> Dear reviewer,
>
> Thank you once again for your time in reviewing our paper and providing valuable feedback. As the discussion period extended till 2nd December, we are reaching out to see if you have any further questions or pending issues.
>
> We have hoped to resolve your concerns with WAR-SAR evaluation metrics, and clarity in writing.
>
> Please let us know if you have any follow-up comments or require additional clarifications.

---

### Official Review · Reviewer_HMAF · 2024-11-04

**Soundness:** 3
**Presentation:** 3
**Contribution:** 3
**Rating:** 8
**Confidence:** 4

**Summary:**

This paper tackles the problem of low-resolution zero-shot classification with vision-language foundation models. The authors observe that zero-shot classification accuracy drops when the resolution of input images decreases. First, they identified limitations in the existing evaluation metrics. They propose an improved relative robustness measure, which avoids misleading high scores when the model behaves closely to random predictions. Additionally, they propose weighted aggregated robustness (WAR) instead of simply averaging scores over datasets, in order to fairly reflect all the datasets. Second, they benchmark 10 foundation models (spanning 66 backbones) using 15 datasets on low-resolution zero-shot classification. Analysis on the dimensions of pretraining data and pretrained model reveals different observations: 1- Robustness to low-resolution correlates positively with pretraining data quality and model size, 2- models with higher resolution input data are less robust to low-resolution, 3- Initial layers are specifically affected when resolution changes. Finally, the authors propose generating high-resolution data using a diffusion model and learning tokens appended to frozen model weight in a teacher-student framework, where the teacher sees high resolution and the student sees both high and low resolutions. Extensive experiments and ablations show the effectiveness of the proposed framework in improving zero-shot classification accuracy, specifically in extreme low-resolution scenarios (e.g., 16x16, 32x32).

**Strengths:**

The paper starts with a good introduction motivating the study of low-resolution (LR) zero-shot classification from a practical angle. The contributions (benchmarking, adjusted metrics, method to improve LR image classification) are clearly conveyed. The potential viability of a solution for LR mispredictions is nicely motivated by Figure 2 in the introduction, showing that the mispredictions in LR case are still semantically decent.

The proposed improvement of the relative robustness score [1] which makes it close to 0 when the predictions are close to random predictions is sound and interesting. Moreover, the proposed weighted aggregated robustness score is well-motivated and shown to better reflect each dataset fairly.

For the benchmarking part, the paper is rich in figures and results showing the problem (degraded LR classification) and insights about correlated factors (model size and pretraining data quality). These results being observed on a wide variety of backbones are interesting and each dimension (for example the pretraining data quality) could be investigated in a specific line of future research. Additionally, pretraining research can benefit from these observations to better account for factors that are expected to improve LR downstream performance.

Finally, proposing a solution based on freezing model weights and appending tokens that are learned in a teacher-student framework is sound and practical. Generating data with a diffusion model to train these tokens is straightforward and shown experimentally to be effective.

[1] Schiappa et al., Robustness Analysis on Foundational Segmentation Models. CVPRW 2024

**Weaknesses:**

The authors mention that low-resolution images are simulated by downsampling high-resolution images using bicubic interpolation. I'm wondering whether using "real" low-resolution images can lead to different observations in all the experimental results. Would the degradation in zero-shot classification accuracy in that case start to be significant from 64x64 or from a higher resolution? Would all the correlations with model size, pretraining dataset size and quality, resolution of pretraining images ...  be the same in this case? Same question for the proposed token learning process.

The phenomenon shown in Figure 2 is interesting: misclassified low-resolution images are still assigned reasonable semantic predictions. Only few examples are shown in this figure. How often does this happen when measured on a full dataset? Is it resulting from the downsampling of the images instead of using images directly captured at low-resolution (relating to first point)?

The LR-TK0 framework is shown in Table 2 to be effective for 16x16 and 32x32. However, for higher resolutions, the metrics (WAR and SAR) and classification accuracy are still on par or lower than the original model. Isn't this compromising the initial model for ''intermediate'' resolutions?

**Questions:**

I am not sure to have understood how results in Figure 7 (right) affect the choice of LR tokens position. Additionally, in Figure 15, what does the x-axis mean? For instance, does [5] mean that LR tokens are introduced starting from the 5th block?

Potential typos:
* Figure 3 (right): $f_{c1}$ $\rightarrow$ $f_{c2}$, $f_{c3}$.
* Figure 4 caption: tradational $\rightarrow$ traditional.
* Figure 5 caption: Pertaining $\rightarrow$ pretraining.
* Line 476: Table 3 $\rightarrow$ Table 4.

.................................................................................................................................................................................................................

Post-rebuttal

.................................................................................................................................................................................................................

I would like to thank the authors for addressing my concerns. I vote with a score of 8 for the paper being the first to benchmark a variety of VLM backbones on low-resolution (LR) images, providing extensive experiments offering insights about the influence of factors like model size and pretraining data quality on the robustness to LR. The identified flaws in existing robustness score and SAR (which measures robustness of a model across datasets) are clear and the proposed solutions are sound (improved relative robustness and WAR, respectively). On the other hand, LR-TK0 as a framework for learning  tokens using pairs of HR-LR images (generated with a diffusion model) is sound and effective especially for extremely low-resolution.
Reading other reviewers' concerns (whom i would like to thank for bringing out some points that helped removing ambiguity in figure captions and details about the proposed metrics), and the corresponding authors' replies, I believe there is no critical weakness in the paper and therefore increase my score.

---

> ### Author Response · Authors · 2024-11-20
> **Rebuttal Response 1**
>
> **W1: The authors mention that.... token learning process.**
>
> Thank you for the insightful remark. To better resolve this issue, we would answer this, in parts:
> **Observation on “real” low-resolution images and degradation at 64 x 64.**
> The reviewer is correct in pointing out that real-world low-resolution images may behave differently from the ones generated synthetically via bicubic interpolation.
> 1) To conduct detailed analysis/benchmarking, we need HR - LR image pairs to measure robustness (drop in accuracy), measure gain in robustness (performance of LR-TK0), and the exact resolution at which accuracy starts dropping (64x64 in this study). To the best of our knowledge, there are no such real-world image classification datasets with cross resolutions or HR-LR pairs. It will be very useful to explore further if such a dataset is available in the future.
> 2) We did a small-scale analysis of real-world low-resolution images, to validate that the models make “semantically reasonable predictions” and “simple mispredictions” on such images `(Figure 29)` and found that the models behave like the synthetic ones `(Figure 2 and Figure 28)`.
> 3) There are works that have seen a correlation between natural distortions and synthetic corruption [1]. Additionally, existing works [2] on low-resolution have traditionally used bicubic interpolation to simulate low-resolution and use it to compare various techniques for improving robustness.
>
> **Analysis and Proposed Technique for Real-world Images**
> Existing real-world benchmarking that overlaps with some of our findings includes:
> 1) Noisy Image Segmentation masks [3]: Preserving original weights helps develop robustness `(LR-TK0)`, and Fine-tuning drops the zero-shot capabilities `(Figure 7 left)`.
> 2) ImageNet-1K pretraining may be more robust than LAION-2B pre-training (in case of fine-tuning). [4] `(Figure 6 left)`.
>
> [1] Michaelis, Claudio, et al. "Benchmarking robustness in object detection: Autonomous driving when winter is coming." arXiv preprint arXiv:1907.07484 (2019).
> [2] P. Li, L. Prieto, D. Mery and P. J. Flynn, "On Low-Resolution Face Recognition in the Wild: Comparisons and New Techniques," in IEEE Transactions on Information Forensics and Security, vol. 14, no. 8, pp. 2000-2012, Aug. 2019, doi: 10.1109/TIFS.2018.2890812.
> [3] Wei-Ting Chen, Yu-Jiet Vong, Sy-Yen Kuo, Sizhou Ma, and Jian Wang. Robustsam: Segment anything robustly on degraded images. 2024.
> [4] Hwang, Jaedong, et al. "ImageNet-RIB Benchmark: Large Pre-Training Datasets Don't Guarantee Robustness after Fine-Tuning." arXiv preprint arXiv:2410.21582 (2024).
>
> -------
> -------
>
> **W2: The phenomenon shown in Figure 2  .... low-resolution (relating to first point)?**
>
> We agree that the reasonable semantic predictions are very interesting and confirm that it occurs very frequently at low resolution. More examples are added in `Supplementary Figure 28`. Real-world low-resolution images are shown in `Figure 29`. which indicates 1) simple mispredictions, 2) semantically correct predictions, and 3) correct predictions.
>
> -------
> -------
>
> **W3: The LR-TK0 framework is .... model for ''intermediate'' resolutions?**
>
> In order to better resolve this issue, we will answer it in parts
> **Par or lower than the original model**
> The reviewer is correct in pointing out that increasing robustness on low resolution comes with a tradeoff drop in performance on higher resolution. In the main paper `[Lines 468–469]`, we discussed that this saying decline in higher-resolution performance is often attributable to a domain shift problem, as highlighted in prior work [1]. For our benchmarking, performance drop starts below 64 x 64, thus have mainly focused on resolutions 32 x 32 and 16 x 16. `[Line 191-193]`.
>
> **Compromising initial performance**
>
> However, 224 x 224 accuracy indicates we are pretty much retaining the initial zero shot accuracy (1-2% accuracy drop). For LR 16 x 16, the accuracy jump is increasing from (4-9%) across different models. It is also noteworthy that other adopted methods in our settings such as Visual Prompt Tuning (VPT) [2] and RobustSAM [3] face similar challenges. Despite this, our framework demonstrates performance closest to the baseline `(Table 4)`,
>
> [1] Efficient Low-Resolution Face Recognition via Bridge Distillation. IEEE Transactions on Image Processing, 2020.
> [2] Visual Prompt Tuning. ECCV 2022.
> [3] RobustSAM: Segment Anything Robustly on Degraded Images. CVPR 2024.

---

> ### Author Response · Authors · 2024-11-20
> **Rebuttal Response 2**
>
> **Q1: I am not sure to have.... choice of LR tokens position.**
>
> Apologies for the confusion, caption `(figure 7)` and `[Line 347-352]` are updated to reflect this:
> `Figure 7` right Layers-wise similarity (e.g. 16 x 16 model layers with 224 x 224 ones). The lower right half indicates the similarity of deeper layers (brighter means more similar), while the upper left represents shallow layers (dull means less similar).
> This means the proposed solution should focus on the initial layers as much as the later final layers. Since HR features are matched with LR features (trainable modules are traditionally introduced after the final features), this inherently gives more weightage to the final layers than the initial layers. Introducing LR tokens at every block treats each layer the same (especially shallow layers). `Figure 15` performance validates this.
>
> -------
> -------
>
> **Q2: Additionally, in Figure 15, ....starting from the 5th block?***
>
> Apologies for the confusion, the caption is updated to reflect this:
> “[i] LR tokens introduced starting from i-th block (\& none after patchification)”, which would mean the LR tokens were introduced starting from the i-th block onwards (5th block in your example). `[Line 521-524]` describes this as well.
>
> -------
> -------
> **Q3: Potential typos:**
> Thank you for bringing these to our attention. All corrections are made to the revised copy (uploaded)
>
> -------
> -------
>
> We will be happy to answer any more doubts and would thank you once again for all the feedback and your time.

---

> ### Comment · Reviewer_HMAF · 2024-11-25
>
> I would like to thank the authors for their detailed reply.
>
> I see the limitation of not having datasets of real LR-HR images pairs. I still believe that the experiments with downsampled images are extensive and that it's reasonable to suppose the conclusions would correlate with natural LR images.
>
> Thanks you for adding more examples in Figure 28. For Figure 29, I'm a bit puzzled about the significance of the observation; I agree the predictions are semantically reasonable, however 4 out of 6 images are well-classified, which means LR isn't a problem for these examples. I believe more interesting examples would be those where the top-1 prediction is incorrect though semantically reasonable. It would be also interesting to report the resolutions of images in Figure 29.
>
> My other concerns are addressed.
>
> I've read other reviewers' comments and i would like to thank them for bringing up details which were clarified by the authors in the revised paper. The paper is heavy in terms of experiments, insights and figures so it's crucial that everything is clearly detailed.
>
> I'm still a bit puzzled about this sentence: "Ideally, the model rankings, after averaging, should stay consistent with individual dataset rankings".
> What does "individual dataset rankings" means? Do the authors mean: the model rankings, after averaging **their accuracy across the datasets**, should stay consistent with **their rankings for individual datasets**?

---

> > ### Author Response · Authors · 2024-11-25
> > **Reply: Reviewer HMAF**
> >
> > We value the reviewer's effort to appreciate the limitations of real-world challenges with LR images analysis, and our rebuttal addressing all the weaknesses W1-3, and Q1-3.
> >
> > Regarding `Figure 29`, we tried to show diversity in predictions not necessarily focusing on Top-1 misprediction. We will add more such examples where Top-1 is mis-correctly predicted in the revised version. The real-world low-resolution (LR) images shown in `Figure 29` are from Google image search and may be already resized to a higher resolution. The resolution for shown images is: Moon-602 × 602, Parking lot-317x210, Deer-309x163, Galaxy-800 × 530, Deer-311 × 162, and Robbery-620 × 349.
> > A simple google search e.g. https://images.app.goo.gl/9gFkfL9D6hLWAhgS9 (500 x 290) reveals how resizing can lead to misleading resolution values.
> >
> > Apologies for the confusion, and you are correct in your understanding, "The model rankings, after averaging their robustness across the datasets, should stay consistent with their rankings for individual datasets.” We will rephrase this according to the reviewers' suggestions and update the revised version `[Line 248-250]`:
> >
> > *“When comparing models, their robustness scores are averaged across datasets (SAR). Ideally, model ranking, after averaging robustness across datasets, should stay consistent with their rankings on individual datasets.”*

---

> ### Comment · Reviewer_HMAF · 2024-11-25
>
> Thanks to the authors for their reply.
>
> I won't linger more on the point about Figure 29. I understand the challenge of real low-resolution and i see how resizing can be misleading.
>
> Remark: I wrote erroneously accuracy instead of robustness in my suggested reformulation (typo). Sorry for that. It's clear in the paper that it's robustness score, sure!
>
> Overall, I believe there is no ''seriously critical'' point in the paper. Given the detailed rebuttal by the authors and the valuable contribution of the paper, I am happy to increase my score from 6 to 8 :)

---

> > ### Author Response · Authors · 2024-11-26
> > **Reply: Reviewer HMAF**
> >
> > We appreciate your quick and prompt reply.
> > Thank you

---

### Author Response · Authors · 2024-11-20
**General Comment addressing all the reviewers**

We thank all the reviewers for the insightful comments and feedback to improve our work. We really appreciate all the constructive criticism and have incorporated all of them in the revised version.
Some of the highlights of our revisions include:

1) We have reworded (simplified) the proposed metrics **WAR** "Problem B) SAR overlooks datasets” `[Line 248-269]`
2) All the captions of the figure are now more descriptive  (d8wv)
3) The revised Supplementary has more examples for `Figure 2`, i.e. model making semantically reasonable predictions at low resolutions `(Figure 28)` and some real-world examples `(Figure 29)` showing correct predictions, wrong predictions, and semantically reasonable predictions.
4) Spacing between the figures is increased for easy viewability.
5) All the typos and corrections are fixed.

We are grateful to the reviewers for acknowledging some of the strengths of our work:
1) Benchmarking 66 foundation models against low resolution can serve as an important future research direction. [TJWB, d8wv].
2) The proposed WAR metric highlights key issues in the existing robustness metrics while representing all the datasets fairly. [ HMAF, TJWB, ]
3) Extensive benchmarking offers a diverse perspective on various aspects of low resolution. [HMAF, pEiF, d8wv, TJWB]
4) LR-TK0 offers an efficient and effective method for improving LR robustness [HMAF, pEiF]

We really appreciate the reviewers for their valuable time and all the feedback for improving our submission. We will be happy to answer any further questions. We have more detailed individual responses for each reviewer and have tried resolving every comment separately.

---

### Public Comment · ~Priyank_Pathak1 · 2025-05-18
**We changed the title of the paper.**

A common pattern we saw during the poster session was the interest in "how to improve robustness of the Foundation Models" (75% of the paper).Yet the original title of the paper didn't reflect that. Hence we decided to change the name of the paper  from

"LR0.FM: Low-Resolution Zero-shot Classification Benchmark For Foundation Models"

to

**"LR0.FM: Low-Res Benchmark and Improving Robustness for Zero-Shot Classification in Foundation Models"**

The bibtex to cite the same would now be :

```
@inproceedings{
pathak2025lrfm,
title={{ LR0.FM: Low-Res Benchmark and Improving Robustness for Zero-Shot Classification in Foundation Models} },
author={Priyank Pathak and Shyam Marjit and Shruti Vyas and Yogesh S Rawat},
booktitle={The Thirteenth International Conference on Learning Representations},
year={2025},
url={https://openreview.net/forum?id=AsFxRSLtqR}
}
```

---

### Meta-Review · Area_Chair_112E · 2024-12-16

**Metareview:**

The reviewers appreciated the well-designed benchmarks with reasonable evaluation metrics and extensive experiments, and the sound and practical solution. They also raised concerns with lack of experiments with real low-resolution images (HMAF), presentation issues like missing details and lack of clarity on the explanation of SAR (pEiF, d8wv), unclear motivation of studying extremely low-resolution image recognition of foundation models (TJWB), lack of comparisons with other relevant benchmarks (TJWB), and unclear connection between the benchmark and the method (TJWB). The authors' rebuttal, revision, and subsequent responses in the discussion period address most of these concerns and suggestions, and consequently, all the reviewers were supportive of the paper after the discussion period. The AC agrees with the reviewers and recommends acceptance. The authors are strongly encouraged to carefully revise the paper to reflect the valuable comments by the reviewers, to add new results brought up in the rebuttal and discussions, and to further improve the quality of writing.

**Additional Comments On Reviewer Discussion:**

After active discussions with the authors, the reviewers unanimously champion the paper. I cannot find any reason to overturn this agreement. Below I summarize the major concerns of the reviewers and how they are resolved.

- **Lack of experiments with real low-resolution images (HMAF)**: The rebuttal with additional results and the revision successfully addressed this issue.
- **Presentation issues (pEiF, d8wv)**: These were the major concerns of the reviewer with the most negative pre-rebuttal score. To be specific, the reviewers raised concerns with figures hard to read, missing details, lack of clarity on the explanation for one of the proposed metric (i.e., SAR), and some issues in captions. These concerns have been well addressed by the revision, and as a result, the reviewers upgraded their scores.
- **Unclear motivation -- "images in 16x16 or 32x32 are not the main target of VL models" (TJWB)**: The authors' response to this comment sounds reasonable and convincing to me, but it seems failed to fully assuage the concern of the reviewer. However, the reviewer believes the paper is valuable enough even with this weakness and very strongly championed the paper.
- **Lack of comparisons with other relevant benchmarks (TJWB)**: Well resolved by the rebuttal.
- **Unclear connection between the benchmark and the method (TJWB)**: The authors' response to this comment sounds convincing, but it seems failed to fully assuage the concern of the reviewer. However, the reviewer believes the paper is valuable enough even with this weakness and very strongly championed the paper.

---

### Decision · Program_Chairs · 2025-01-22

Accept (Poster)